# Selective androgen receptor degrader (SARD) to overcome antiandrogen resistance in castration-resistant prostate cancer

Meng Wu[1,2,3†], Rongyu Zhang[4†], Zixiong Zhang[1†], Ning Zhang[1], Chenfan Li[4], Yongli Xie[1], Haoran Xia[2], Fangjiao Huang[4], Ruoying Zhang[4], Ming Liu[2], Xiaoyu Li[1]*, Shan Cen[1]*, Jinming Zhou[1,4]*

[1]Institute of Medicinal Biotechnology, Chinese Academy of Medical Sciences, Beijing, China; [2]Department of Urology, Beijing Hospital, National Center of Gerontology, Institute of Geriatric Medicine, Chinese Academy of Medical Sciences, Beijing, China; [3]Department of Medical Research Center, State Key Laboratory of Complex Severe and Rare Diseases, Peking Union Medical College Hospital, Chinese Academy of Medical Sciences & Peking Union Medical College, Beijing, China; [4]Key Laboratory of the Ministry of Education for Advanced Catalysis Materials, Department of Chemistry, Zhejiang Normal University, Jinhua, China

*For correspondence:
lixiaoyu@imb.pumc.edu.cn (XL);
shancen@imb.pumc.edu.cn (SC);
zhoujinming@zjnu.edu.cn (JZ)

†These authors contributed equally to this work

**Abstract** In patients with castration-resistant prostate cancer (CRPC), clinical resistances such as androgen receptor (AR) mutation, AR overexpression, and AR splice variants (ARVs) limit the effectiveness of second-generation antiandrogens (SGAs). Several strategies have been implemented to develop novel antiandrogens to circumvent the occurring resistance. Here, we found and identified a bifunctional small molecule Z15, which is both an effective AR antagonist and a selective AR degrader. Z15 could directly interact with the ligand-binding domain (LBD) and activation function-1 region of AR, and promote AR degradation through the proteasome pathway. In vitro and in vivo studies showed that Z15 efficiently suppressed AR, AR mutants and ARVs transcription activity, downregulated mRNA and protein levels of AR downstream target genes, thereby overcoming AR LBD mutations, AR amplification, and ARVs-induced SGAs resistance in CRPC. In conclusion, our data illustrate the synergistic importance of AR antagonism and degradation in advanced prostate cancer treatment.

## Editor's evaluation

The present study reports the discovery and preclinical evaluation of a novel therapeutic agent for the treatment of castration-resistance prostate cancer through inducing degradation of androgen receptor. The major strength of this study is the identification of a novel lead compound and its interesting in vitro and in vivo activities in prostate cancer models.

## Introduction

Prostate cancer (PCa) is one of the most common cancers and the second leading cause of cancer-related death for men in western countries (*Siegel et al., 2022*; *Sung et al., 2021*). Advanced PCa initially responds to androgen deprivation therapy (ADT), but invariably fails and recurs as lethal castration-resistant prostate cancer (CRPC) (*Harris et al., 2009*; *Desai et al., 2021*). Androgen

receptor (AR) signaling plays a crucial role in the progress and survival of CRPC (*Dai et al., 2017*). Second-generation antiandrogens (SGAs), such as enzalutamide (ENZa), abiraterone, apalutamide, and darolutamide, improve the overall survival time and decline prostate-specific antigen (PSA) levels in patients with CRPC (*Sternberg et al., 2020*; *Armstrong et al., 2019*; *Smith et al., 2021*; *Smith et al., 2022*; *Ryan et al., 2015*; *de Bono et al., 2011*). Despite the initial benefit of these agents, their success in treating CRPC has been eliminated by the emergence of drug resistance. Multiple possible mechanisms for the development of drug resistance have thus far been identified, including mutations in the AR LBD, amplification of AR, expression of AR splice variants (ARVs), and intra-tumoral de novo androgen synthesis (*Buttigliero et al., 2015*; *Robinson et al., 2015*; *Karantanos et al., 2015*). Therefore, more effective therapies are urgently required to conquer the SGAs drug resistance.

Several strategies have been implemented to develop novel antiandrogens to circumvent the occurring resistance. The first strategy is to develop new competitive antiandrogens targeting the AR hormone-binding pocket (HBP) site, such as darolutamide (*Smith et al., 2022*). Another strategy is to target the AR signaling axis beyond the HBP site, which includes activation function-1 (AF1), activation function-2, binding function 3, and the DNA binding site through active compounds, such as EPI-001, VPC-14449 (*Caboni and Lloyd, 2013*). Recently, down-regulating both AR protein and AR mRNA levels has attracted attention due to their potential in the discovery and development of new antiandrogens. The most exciting progress is AR degradation based on the proteolysis targeting chimeras (PROTACs) concept, with various of these AR PROTACs developed with a $DC_{50}$ (drug concentration that results in 50% protein degradation) potency up to 1 nM. However, low cell permeability, poor pharmacokinetic properties, and complex chemical structures may restrict the clinical application of PROTAC drugs (*He et al., 2020*). What's more, LBD-targeted AR PROTACs cannot degrade ARVs which were associated with unfavorable clinical outcomes in patients with CRPC (*Fettke et al., 2020*).

The selective estrogen receptor degrader fulvestrant approved by the FDA in 2002 expanded treatment choices for advanced breast cancer (*Bross et al., 2003*), which gave rise to next-generation novel degraders with promising antitumor activity in recent years (*Nardone et al., 2019*). *Bradbury et al., 2011* suggested that similar specific downregulation or degradation of AR might be proved beneficial in the treatment of CRPC. Therefore, selective AR degraders (SARDs) which could synthetically degrade and antagonize AR may be an efficient strategy to overcome the drug resistance in the antiandrogen therapy of CRPC. Based on structural modification of the AR antagonists and the tissue-selective AR agonist enobosarm, Miller et al. designed a series of SARDs, namely UT-155, UT-69, and UT-34, which could induce AR ubiquitin-proteasome degradation *via* binding to AF-1 of the AR to reduce its stability (*Ponnusamy et al., 2017*; *Ponnusamy et al., 2019*; *Hwang et al., 2019*). Notably, the degradation potency of these compounds for ARVs is quite limited. In the present study, we determined that Z15 screened by rational drug design as an AR antagonist and degrader via direct binding to the AR LBD and AR AF1, could overcome AR LBD mutations, AR amplification, and ARVs-induced SGAs resistance of CRPC in vitro and in vivo.

## Results

### Identifying Z15 as an AR inhibitor

To develop novel AR inhibitors and overcome antiandrogen resistance, we previously constructed a common molecular characteristic pharmacophore model, and screened ~7.5 million compounds from the ZINC lead-like database and ChemDiv database. About 47,202 compounds matched more than four features of the filtering model. Next, these compounds were docked into the HBP of the antagonistic AR. Then, compounds with the top 1000 docking scores were chosen for ADMET prediction by Discovery Studio v3.5. Finally, 80 hits with high drug-likeness were selected and purchased for further bioactivity evaluation (*Figure 1—figure supplement 1* and *Supplementary file 1a*).

To preliminarily evaluate the influences for AR transcriptional activity of these 80 candidates, human prostate cancer cells PC-3 co-transfected with wild-type AR (wt-AR) and PSA-luc were incubated with 5α-dihydrotestosterone (DHT) and 10 μM candidate compounds for 24 h. The cell lysates were collected and AR transcriptional activity was detected by dual-luciferase reporter assay. We identified 19 compounds that showed more than 25% AR transcription inhibition activity, among which compound Z15 (structure shown in *Figure 1A*) exhibited the most potent AR inhibition activity

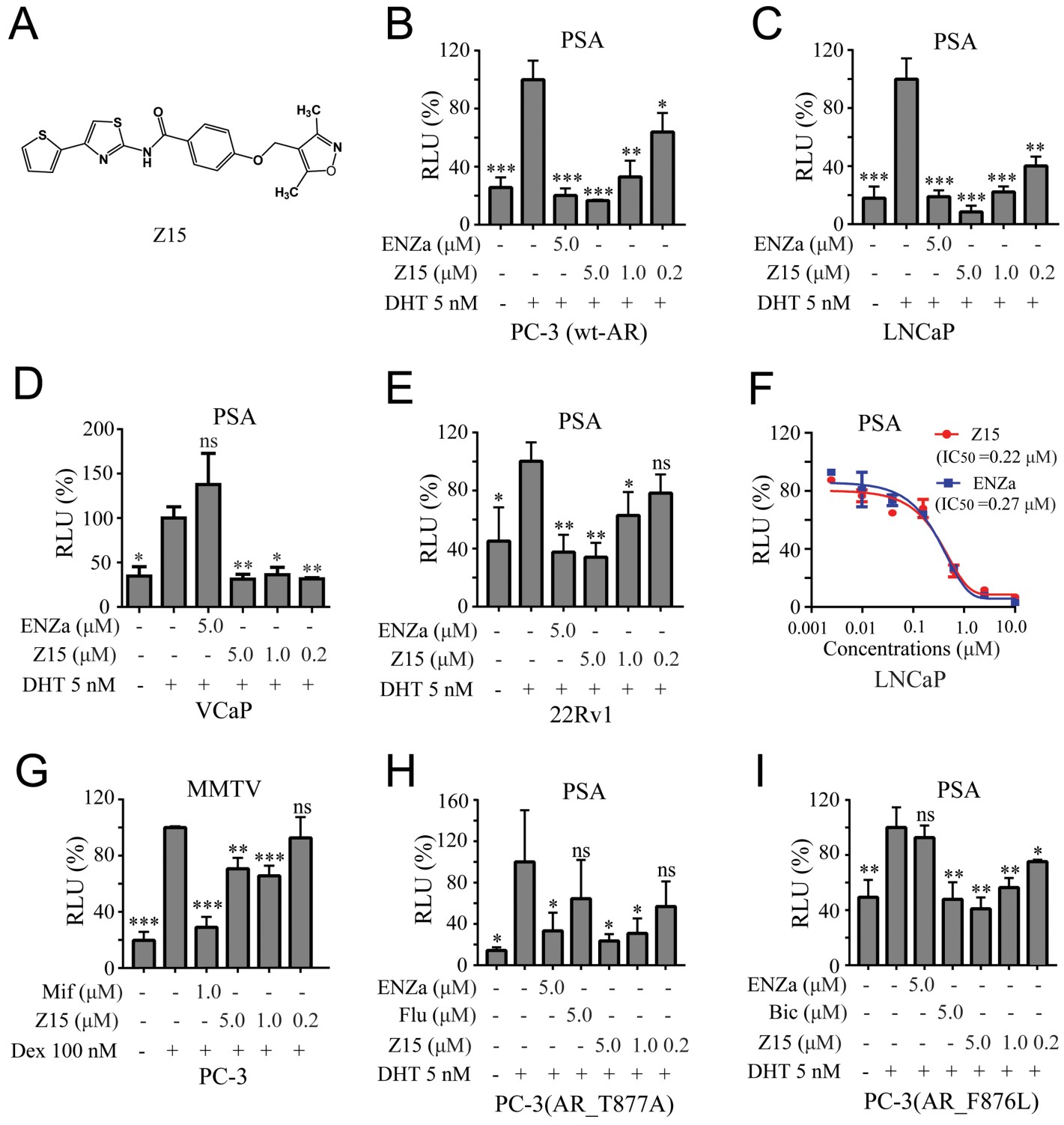

**Figure 1.** Z15 specifically inhibits the transcription activity of AR and AR mutants. (**A**) Chemical structure of Z15. (**B**) Dual-luciferase reporter assay to measure PSA-luc reporter luciferase activities in PC-3 cells co-transfected with Renilla, AR, and PSA promoter expression vector plasmids, stimulated by 5 nM DHT, and treated with different concentrations of Z15 for 24 hr. (**C**) LNCaP, (**D**) VCaP, (**E**) and 22Rv1 cells co-transfected with Renilla and PSA promoter expression vector plasmids, stimulated by 5 nM DHT, and treated with different concentrations of Z15 for 24 h. (**F**) Dual-luciferase reporter assay to measure PSA-luc reporter luciferase activities in LNCaP cells stimulated by 5 nM DHT, and treated with different concentrations of Z15 or ENZa for 24 hr. (**G**) Dual-luciferase reporter assays to measure MMTV-luc reporter luciferase activities in PC-3 cells co-transfected with Renilla and MMTV promoter expression vector plasmids stimulated by 100 nM Dex, and treated with different concentrations of Z15 for 24 hr. (**H**) Dual-luciferase reporter

*Figure 1 continued on next page*

*Figure 1 continued*

assays to measure PSA-luc reporter luciferase activities in PC-3 cells co-transfected with Renilla, AR_T877A mutation, and PSA promoter expression vector plasmids stimulated by 5 nM DHT treated with different concentrations of Z15 for 24 hr. (**I**) PC-3 cells co-transfected with Renilla, AR_F876L mutation, and PSA promoter expression vector plasmids, treated with different concentrations of Z15 for 24 hr. All experiments were performed in triplicate. Results are shown as mean ± sd. *p<0.05, **p<0.01, ***p<0.001 vs DHT or Dex group. ENZa, enzalutamide; DHT, dihydrotestosterone; Dex, dexamethasone; Mif, mifepristone.

The online version of this article includes the following figure supplement(s) for figure 1:

**Figure supplement 1.** In silico screening procedure.

**Figure supplement 2.** Primary bioactivity evaluation of virtual screened AR inhibitor candidates.

**Figure supplement 3.** The procedure for the synthesis of compound Z15.

**Figure supplement 4.** Z15 specifically inhibits the transcriptional activity of AR rather than GR, ER and PR.

(*Figure 1—figure supplement 2A*). Nevertheless, the glucocorticoid receptor (GR) transcription inhibition activity of Z15 was quite feeble (*Figure 1—figure supplement 2B–C*).

## Z15 selectively suppresses AR and AR mutant transcriptional activity

To further investigate the AR inhibition potency of Z15, we optimized the synthesis route and prepared a sufficient amount of Z15 (*Figure 1—figure supplement 3*). Next, we performed a dual-luciferase reporter assay in several human PCa cell lines including wt-AR-transfected PC-3 and LNCaP cells. The results indicated that Z15 could inhibit DHT-induced transcriptional activities of both exogenous and endogenous AR in a dose-dependent manner (*Figure 1B–C*). Unexpectedly, Z15 showed potent AR transcription inhibition activity in AR overexpression and ENZa-insensitive VCaP cells (*Figure 1D*). In another ENZa resistance 22Rv1 cells which naturally express AR and ARV7, Z15 also inhibited DHT-activated AR transcriptional activity (*Figure 1E*). Moreover, the AR transcription inhibition $IC_{50}$ (half-maximal inhibitory concentration) of Z15 in LNCaP cells was ~0.22 µM, which was comparable to ENZa (*Figure 1F*).

We further detected the selectivity of Z15 in GR-positive PC-3 cells, the results indicated that Z15 hardly inhibited dexamethasone activated GR transcriptional activity compared to the GR antagonist mifepristone (*Figure 1G*). Then, we compared AR, GR, estrogen receptor (ER), and progesterone receptor (PR) transcription inhibition activities of Z15 by dual-luciferase reporter assay. The transcription inhibition $IC_{50}$ of Z15 was 0.41 µM for AR (*Figure 1—figure supplement 4A*), over 20 µM for GR and ER (*Figure 1—figure supplement 4B–C*), and 9.29 µM for PR (*Figure 1—figure supplement 4D*), which suggests that Z15 is a highly selective AR inhibitor.

AR LBD point mutations such as AR T877A (a flutamide-resistant mutation) and AR F876L (ENZa- and apalutamide-resistant mutation), are key causes leading to antiandrogen resistance. Dual-luciferase reporter assay results indicated Z15 could efficiently inhibit DHT-induced both AR T877A and AR F876L transcriptional activities (*Figure 1H–I*). Taken together, these data illustrate Z15 as a potent selective AR inhibitor both for wild-type and mutated ARs.

## Z15 inhibits the AR pathway

Next, we assessed the influence of Z15 on LNCaP cells transcriptome by RNA-sequencing analysis. Obviously, Z15 dose-dependently inhibited a series of DHT-activated AR downstream genes (*Figure 2A*). Then, we detected three canonical AR downstream-regulated genes (*PSA*, *PMEPA1*, and *TMPRSS2*) by quantitative real-time PCR (qRT-PCR) assay. The results revealed that Z15 significantly inhibited the mRNA expression levels of these genes (*Figure 2B*), consistent with the findings of RNA-sequencing. Furthermore, Z15 also decreased DHT-induced *PSA* mRNA levels in the antiandrogen resistance 22Rv1 and VCaP cells (*Figure 2C*).

We further detected the influence of Z15 on AR and PSA protein levels in LNCaP cells. As demonstrated in *Figure 2D*, Z15 reduced DHT-activated PSA protein levels significantly, which was in line with the qRT-PCR analysis. Surprisingly, AR protein levels were also downregulated by Z15, quite different from the effects of ENZa (*Figure 2—figure supplement 1A–B*). Notably, Z15 potently inhibited PSA and AR protein levels in ENZa resistance 22Rv1 and VCaP cells (*Figure 2E–F* and *Figure 2—figure supplement 1C–G*). Then, we evaluated the AR $DC_{50}$ of Z15 in LNCaP and 22Rv1 cells. The AR $DC_{50}$ of Z15 in LNCaP cells was 1.05 µM (*Figure 2G* and *Figure 2—figure supplement 1H*), while in

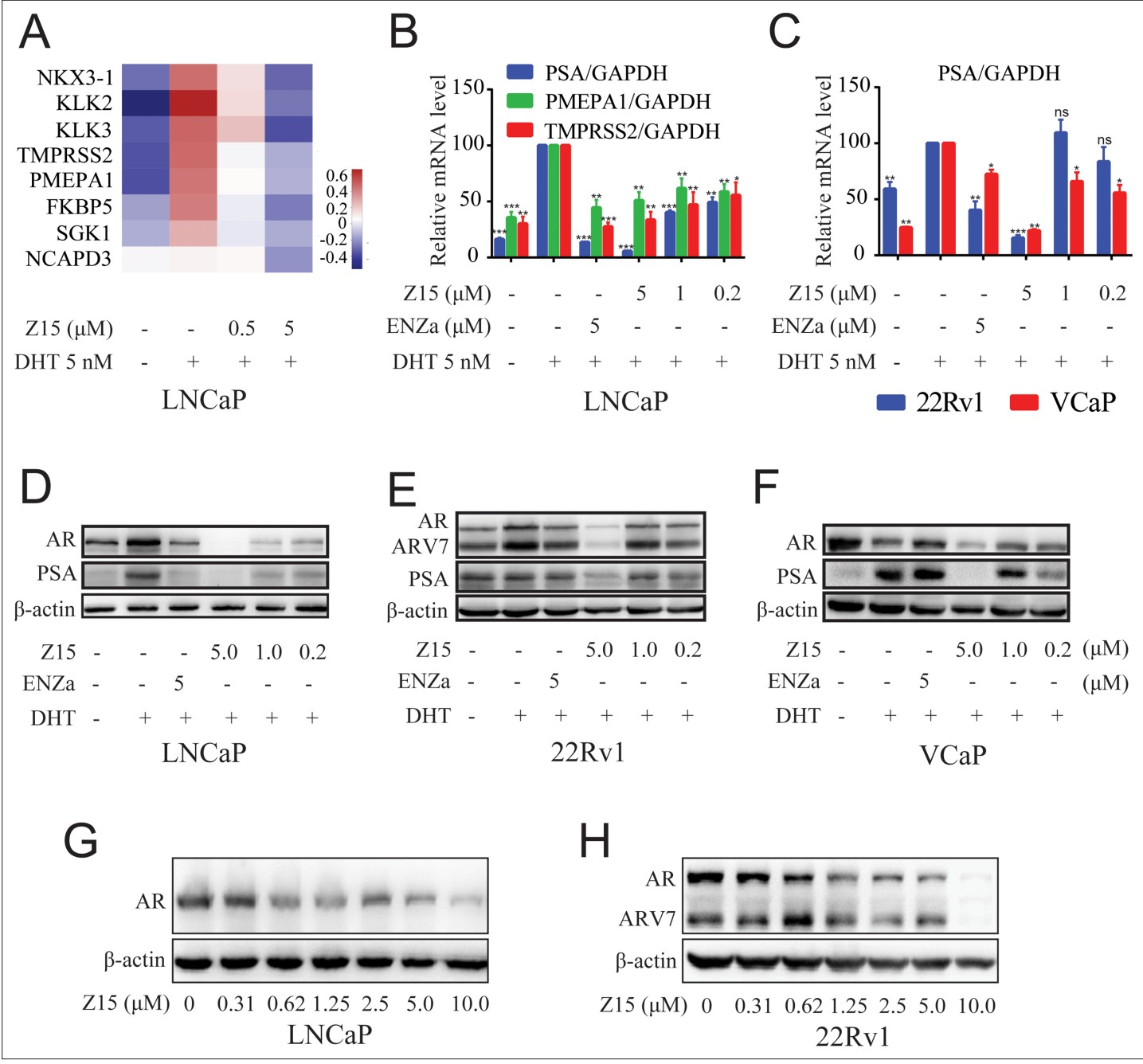

**Figure 2.** Z15 downregulates AR target genes and ARlevels. (**A**) LNCaP cells treated with vehicle, 0.5, or 5 μM Z15 in the presence of 5 nM DHT for 24 hr before performing RNA-sequencing. Heatmap shows the expression levels of AR target genes. (**B**) The mRNA levels of *PSA*, *PMEPA1*, and *TMPRSS2* measured by quantitative-PCR and normalized to GAPDH in LNCaP cells treated with vehicle or different concentrations of Z15 in the presence of 5 nM DHT for 24 hr. (**C**) The mRNA levels of *PSA* measured by quantitative-PCR and normalized to GAPDH in 22Rv1 and VCaP cells treated with vehicle or different concentrations of Z15 in the presence of 5 nM DHT for 24 hr. (**D**) Western blot analysis of LNCaP cells treated with indicated concentrations of Z15 in the presence of 5 nM DHT for 24 hr, before cell lysing and determining PSA and AR protein levels. (**E**) Western blot analysis performed in 22Rv1 cells. (**F**) Western blot analysis performed in VCaP cells. (**G**) Western blot analysis of LNCaP cells treated with indicated concentrations of Z15 in the absence of DHT for 24 hr, before cell lysing, and determining AR protein levels. (**H**) Western blot analysis of 22Rv1 cells treated with indicated concentrations of Z15 in the absence of DHT for 24 hr, before cell lysing, and determining AR protein levels. Experiments were performed in triplicate. Results are shown as mean ± sd. *p<0.05, **p<0.01, ***p<0.001 vs DHT group.

The online version of this article includes the following figure supplement(s) for figure 2:

**Figure supplement 1.** Z15 down regulates AR and ARV7 protein levels.

*Figure 2 continued on next page*

*Figure 2 continued*

**Figure supplement 2.** The influence of Z15 and ARV-110 on LNCaP cells global proteomics.

**Figure supplement 3.** Z15 have no influence on GR, HSP90, and CDK7 protein levels.

22Rv1 cells it was 1.16 μM and the ARV7 $DC_{50}$ was 2.24 μM (*Figure 2H* and *Figure 2—figure supplement 1I*).

In addition, we performed a 4D-label free proteomics study to analyze the effect of Z15 on global protein levels in LNCaP cells. Among 5334 quantifiable proteins, AR LBD-targeted PROTAC molecule ARV-110 significantly reduced 34 proteins and Z15 downregulated 69 proteins compared to the DHT group (*Figure 2—figure supplement 2* **file 1d-e**). Both Z15 and ARV-110 reduced AR, KLK3, and TMPRSS2 protein levels significantly (*Figure 2—figure supplement 2A–B*). KEGG analysis also proved that these two compounds had a similar influence on the functional pathways (*Figure 2—figure supplement 2C–D*). Additionally, to verify the specificity of Z15 downregulated AR protein levels, we chose 3 AR pathway related but independent proteins GR, HSP90 (AR chaperonin), and cyclin-dependent kinases 7 (CDK7) as controls. Western blot analysis indicated that Z15 has no influence on GR, HSP90, and CDK7 protein levels in 22Rv1 cells (*Figure 2—figure supplement 3*). Collectively, these data suggest that Z15 is a novel specific AR pathway inhibitor, which may play a role as an AR antagonist as well as an AR and ARV7 degrader.

## Z15 inhibits DHT-induced AR nuclear translocation

Androgen-binding initiates AR activation, induces its conformational change, and reveals the nuclear localization signal of AR. The hormone-bound AR dimerizes and translocates to the nucleus, where it binds to DNA and interacts with a series of transcriptional coregulators to regulate target gene expression. Accordingly, we investigated whether Z15 disturbed androgen-induced AR nuclear translocation. As shown in *Figure 3A–B*, the DHT treatment could promote the importing of AR into the nuclear compared to untreated group, while both ENZa and Z15 blocked DHT-induced AR nuclear translocation. This result proves that Z15 can inhibit DHT-induced AR nuclear translocation.

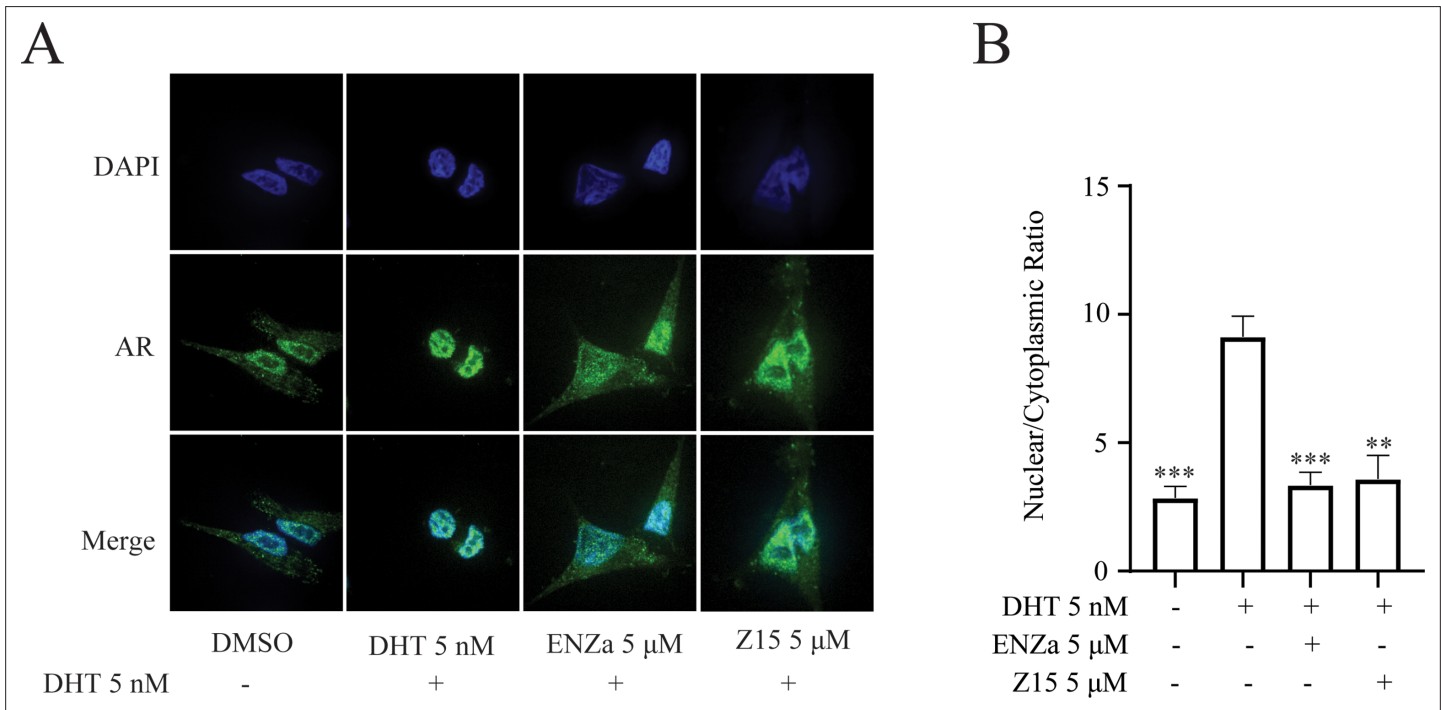

**Figure 3.** Z15 inhibits AR nuclear localization. (**A**) Nuclear localization of AR in LNCaP cells treated with vehicle or 5 μM compounds in the presence of 5 nM DHT for 4 h. (**B**) Quantitative analysis of AR nuclear localization. Experiments were performed in triplicate.

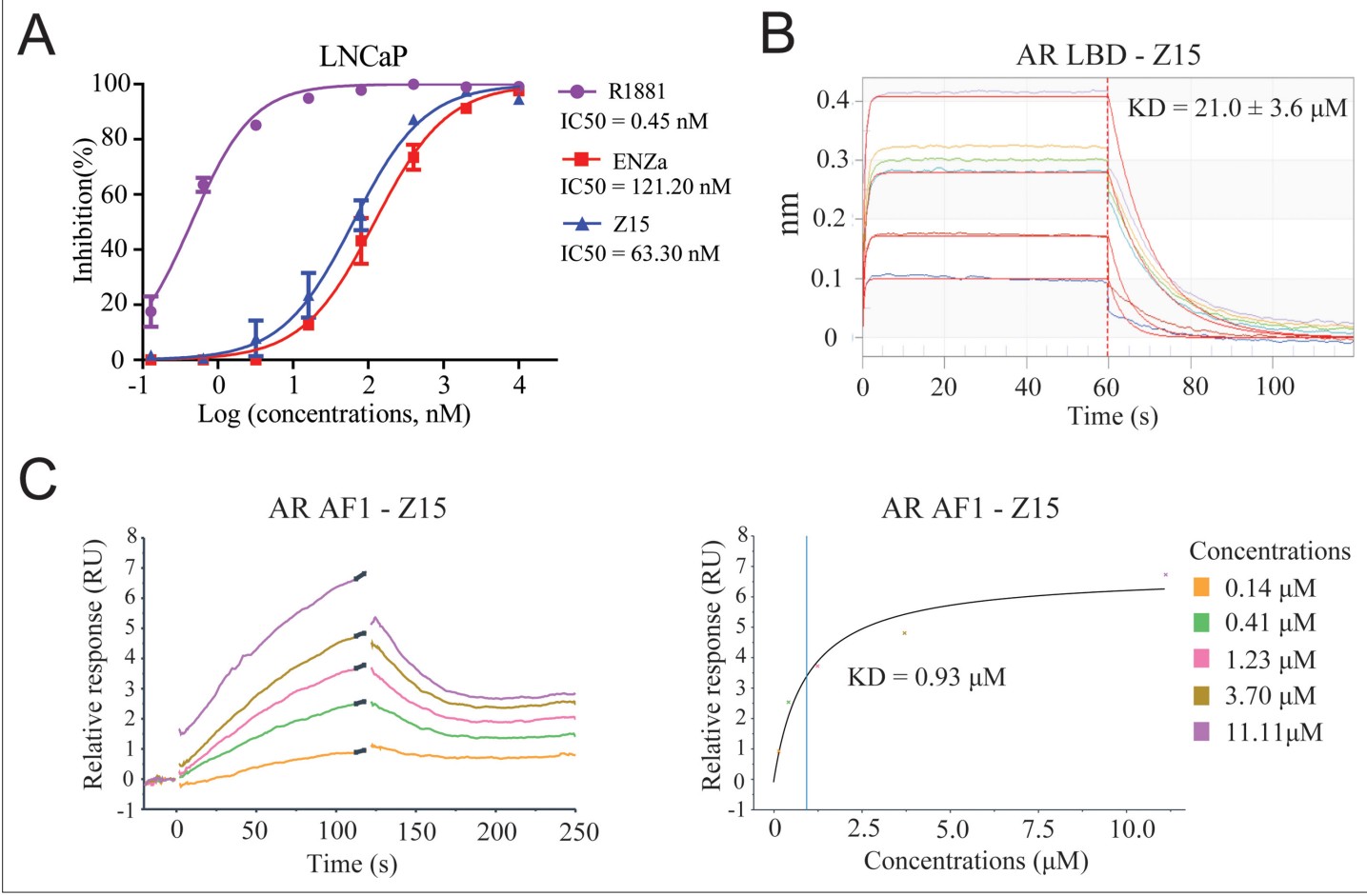

**Figure 4.** Z15 directly binds to AR. (**A**) Competitive binding assay to detect binding affinity of R1881, ENZa, and Z15 to AR LBD, 1 nM radioligand [³H] DHT and LNCaP cytosol were used. (**B**) Biolayer interferometry measurements of Z15 binding to AR LBD. (**C**) Sensorgram and steady state fitted results of surface plasmon resonance assay to detect binding affinity between Z15 and AF1. Experiments were performed in triplicate.

The online version of this article includes the following figure supplement(s) for figure 4:

**Figure supplement 1.** Competitive binding of Z15 to AR-LBD evaluated by AR fluorescence polarization (FP) assay.

**Figure supplement 2.** Z15 could bind to AR LBD and AR AF1 directly.

**Figure supplement 3.** The influences of ARV-110 and UT-34 on 22Rv1 AR and ARV7 protein levels.

## Z15 binds directly to AR LBD and AR AF1

Since the chemical structure of Z15 is remarkably different from that of previously reported AR antagonists, we next evaluated whether Z15 directly binds to AR in a similar manner as ENZa. The AR competitive binding assay was performed to demonstrate the direct interaction between Z15 and AR, whereby compounds in competition with the radioligand [³H] DHT in cytosolic lysates from LNCaP cells were measured. Synthetic androgen R1881 displayed strong binding potency to AR with an $IC_{50}$ value of 0.45 nM, which indicated the feasibility of this assay system. The binding affinity between ENZa and AR was 121.2 nM. Interestingly, Z15 showed a comparable binding affinity to ENZa, with an $IC_{50}$ value of 63.3 nM (*Figure 4A*). In addition, our fluorescence polarization assay demonstrated Z15 could compete with androgen binding to AR LBD (*Figure 4—figure supplement 1*). Besides, the biolayer interferometry (BLI) measurement also revealed that both ENZa and Z15 possess AR LBD binding ability (*Figure 4B*, *Figure 4—figure supplement 2A*). These data suggested that Z15 could antagonize AR by directly targeting the LBD region. AR LBD targeted compound ARV-110 has been shown as an efficient AR degrader in preclinical research, however, it could not induce ARV7 degradation in 22Rv1 cells (*Figure 4—figure supplement 3A–C*). Since Z15 could degrade both AR and ARV7, we wondered if Z15 could also bind to other regions of AR to induce ARV7 degradation.

Hence, we investigated the binding affinity between Z15 and AR AF1, as AF1 is an important drug target region of AR. The surface plasmon resonance assay indicated that Z15 could directly bind to AR AF1 with a KD value of 0.93 μM (*Figure 4C*). Z15 was also detected to potently bind to AR AF1 with a comparable binding affinity to AR AF1 inhibitor UT-34 by BLI assay (*Figure 4—figure supplement 2B–C*). Unexpectedly, UT-34 could not induce ARV7 degradation in 22Rv1 cells from western blot analysis (*Figure 4—figure supplement 3D–F*). As a control, we did not find any binding potency between AR AF1 and ENZa even at 200 μM (*Figure 4—figure supplement 2D*). These data illustrate that Z15 potently inhibits ARV7 by directly binding to AR AF1.

## Z15 promotes AR degradation through the proteasome pathway

We have shown that Z15 could reduce AR and ARV7 protein levels and conjectured that it is an AR degrader. To confirm this hypothesis, we detected the influence of Z15 on AR protein and mRNA levels in LNCaP cells without DHT treatment. Certainly, Z15 reduced AR protein levels in a dose-dependent manner without influencing the AR mRNA levels (*Figure 5A* and *Figure 5—figure supplement 1A*). Moreover, we observed similar effects of Z15 on AR protein and mRNA levels in ENZa resistance cell lines 22Rv1 (*Figure 5B* and *Figure 5—figure supplement 1B–C*) and VCaP cells (*Figure 5C* and *Figure 5—figure supplement 1D*). Western blot analysis for AR in LNCaP cells treated with protein synthesis inhibitor cycloheximide, showed that Z15 accelerated AR degradation (*Figure 5D* and *Figure 5—figure supplement 1E*). These data indicate that Z15 is indeed an AR degrader.

The ubiquitin-proteasome pathway (UPP) is the main participant that regulates intracellular protein degradation. To explore whether Z15 promoted AR degradation through UPP, LNCaP cells were treated with Z15 in the presence or absence of proteasome inhibitor MG132. Indeed, Z15 reduced the AR protein levels after 8 hr treatment, while AR protein levels reduction was counteracted by MG132. Similarly, Z15 induced AR protein decline was also counteracted by MG132 in VCaP cells (*Figure 5E* and *Figure 5—figure supplement 1F–G*). Furthermore, Z15 treatment strikingly induced ubiquitination of AR (*Figure 5F*). Together, these results indicate that Z15 degrades AR through the UPP.

## Z15 inhibits proliferation and induces apoptosis in AR-positive CRPC cell lines

As Z15 exhibited clear AR and ARV7 inhibition and degradation potency, we next investigated the effects of Z15 on cell proliferation activity in AR-positive CRPC cell lines VCaP and 22Rv1 cells. In VCaP cells, Z15 (IC$_{50}$=1.37 μM) showed comparable proliferation inhibition potency with positive control ARV-110 (IC$_{50}$=0.86 μM). However, in ARV7-positive 22Rv1 cells, the proliferation inhibition activity of Z15 (IC$_{50}$=3.63 μM) was much stronger than that of ARV-110 (IC$_{50}$=14.85 μM). Both Z15 and ARV-110 displayed weak inhibition effects on the proliferation activity of AR-negative PC-3 and DU145 cells (*Figure 6A*). To estimate the effects of Z15 on CRPC cell clonogenic activity, we exposed 22Rv1 and PC-3 cells to 1 μM Z15 or ARV-110 for 2 weeks. As a result, Z15 significantly decreased the 22Rv1 cell colony numbers compared to both negative control and ARV-110, whereas both Z15 and ARV-110 showed no influence on the PC-3 cell colony numbers (*Figure 6B*). Thus, we proved that through blocking and degrading AR, Z15 could selectively inhibit the proliferation of AR and ARV7 positive CRPC cell lines. Furthermore, based on PCa patient tumor derived tissues, we cultured PCa organoids and treated the organoids with 1 μM Z15 for 7 days. The results indicated that Z15 significantly inhibited PCa organoids proliferation compared to the control group (*Figure 6C*). What's more, western blot analysis indicated that Z15 also promoted the apoptosis of VCaP and 22Rv1 cells in a dose-dependent manner and time-dependent manner, while Z15 showed no influence on the apoptosis of AR negative DU145 cells (*Figure 6D–E* and *Figure 6—figure supplement 1A–E*).

## Z15 inhibits CRPC xenografts growth

Our previous experiments proved that Z15 is a selective AR degrader and antagonist with excellent anti-PCa activity in vitro. To evaluate the PCa inhibition activity of Z15 in vivo, we established subcutaneous xenograft tumor models of 22Rv1 cells in the flanks of male BALB/c nude mice. When tumor volumes reached an average size of 50~100 mm³, animals were treated with vehicle control, 10 mg/kg Z15, or 20 mg/kg Z15 for 21 days intragastrically once daily. There were no measurable side effects observed in Z15-treated mice as assessed by the mice body weight (*Figure 7A*). Treatment of mice with 10 and 20 mg/kg Z15 both suppressed 22Rv1 tumor progression and decreased the

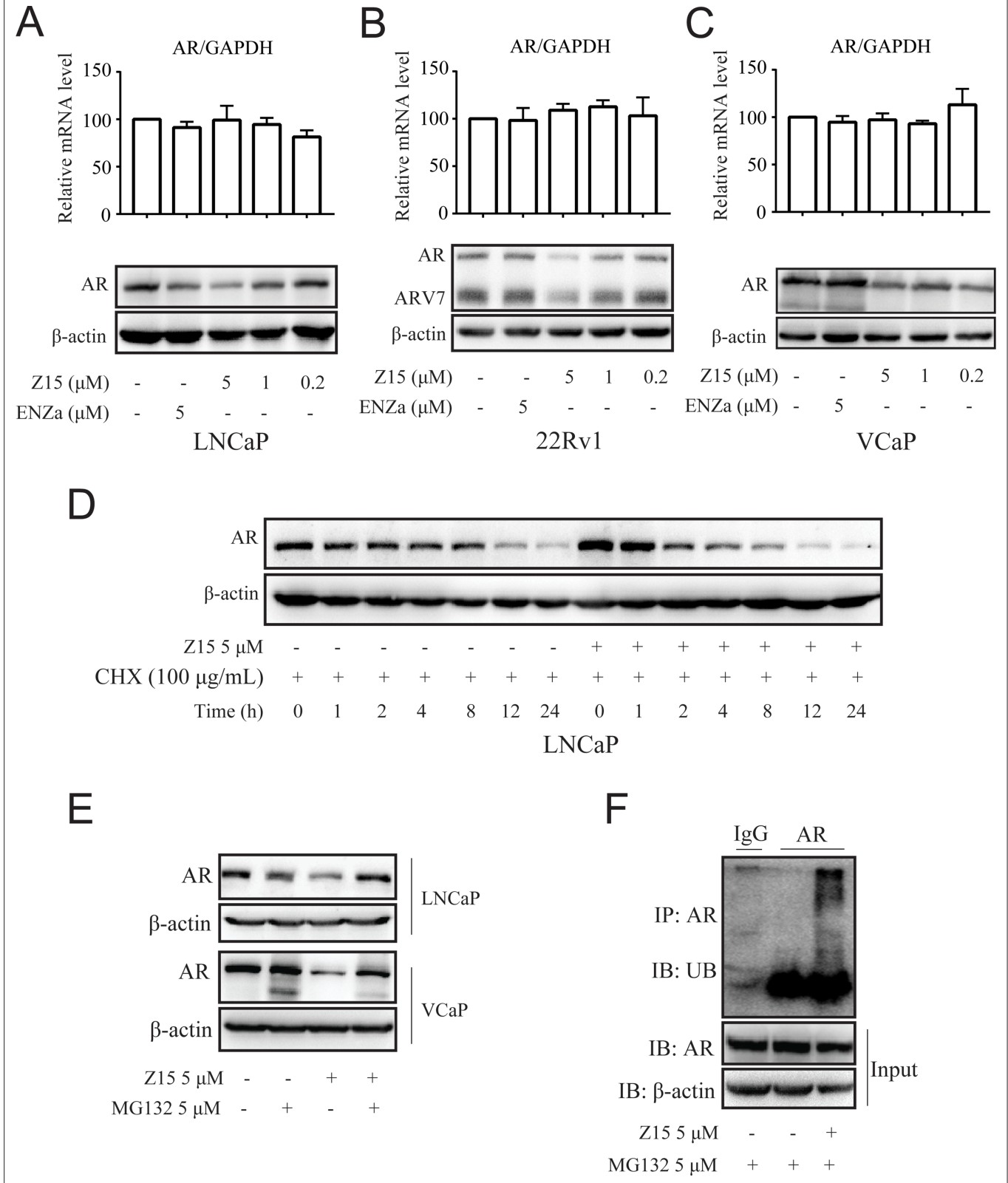

**Figure 5.** Z15 promotes AR degradation in proteasome pathway-dependent manner. A-C Western blot analysis of AR protein levels, and quantitative-PCR normalized to GAPDH of AR mRNA levels in LNCaP (**A**), 22Rv1 (**B**), and VCaP (**C**) cells treated with indicated concentrations of Z15 in the absence of DHT for 24 hr. (**D**) Western blot analysis of AR in LNCaP cells treated with 100 μg/mL CHX in the presence or absence of 5 μM Z15 for indicated time points. (**E**) Western blot analysis of AR protein levels in LNCaP and VCaP cells treated with 5 μM Z15 or/and 5 μM MG 132 for 8 hr.

*Figure 5 continued on next page*

*Figure 5 continued*

(**F**) Immunoprecipitation done using anti-AR and immunoblotting with anti-Myc antibody in 22Rv1 cells co-transfected with Myc-tag CW7-UB plasmids treated with or without 5 µM Z15 in the presence of 5 µM Mg132 for 12 hr. Input: immunoblot of lysates probed with AR antibody. Experiments were performed in triplicate. All results are shown as mean ± sd. CHX, cycloheximide.

The online version of this article includes the following figure supplement(s) for figure 5:

**Figure supplement 1.** Z15 promotes AR degradation in proteasome pathway-dependent manner.

tumor weight significantly (*Figure 7B–C*). In addition, western blot analysis indicated that AR, ARV7, and PSA protein levels in the tumor tissues were significantly declined in both 10 and 20 mg/kg Z15 treatment groups (*Figure 7D*, *Figure 7—figure supplement 1A–C*). Immunohistochemistry analysis also revealed that Z15 reduced the Ki-67 and PSA protein levels in tumor tissues (*Figure 7E*). Taken together, our data indicate that Z15 could inhibit the growth of CRPC both in vitro and in vivo.

### Identifying Z15 analogs as AR inhibitor

Since Z15 showed inspiring CRPC inhibition potency, we searched the SciFinder and ZINC database to seek Z15 chemical structure analogs. Seven compounds (*Figure 8—figure supplement 1*, *Supplementary file 1c*) with more than 80% similarity to Z15 were eventually obtained for further bioactivity evaluation. Dual-luciferase reporter assay indicated that most of these compounds pronouncedly inhibited DHT-activated AR transcriptional activity at 1 µM except for ZL-1 (*Figure 8—figure supplement 2A*), suggesting that the dimethyl isoxazole group plays an indispensable role in the AR inhibitory activity of Z15 and its analogs. Western blot analysis revealed that these active molecules also reduced AR and DHT-induced PSA protein levels (*Figure 8—figure supplement 2B*). Furthermore, we detected the AR transcription inhibition $IC_{50}$ of six active Z15 analogs. Among these molecules, ZL-2 and ZL-4 exhibited the strongest AR transcription inhibition potency, while others also showed comparable AR inhibition activity compared to Z15 (*Figure 8A–B*). Western blot analysis revealed that these active molecules could reduce AR and PSA protein levels in a dose-dependent manner. Notably, some of these compounds showed stronger AR downregulation activity than Z15 (*Figure 8C* and *Figure 8—figure supplement 3A–D*). Together, these results indicate that through chemical structural modification to Z15, more and more selective AR degraders with stronger AR inhibition activity might be found in the near future.

## Discussion

SGAs are becoming more prominent in the clinical treatment of patients with CRPC. However, drug resistance caused by AR mutation, AR amplification, and ARVs, has been widely reported to restrict the clinical benefits of these therapies (*Buttigliero et al., 2015*; *Robinson et al., 2015*). AR remains a crucial target for CRPC therapeutic development because of its key function in the progress of CRPC. In this study, we identified a compound Z15 that selectively inhibited AR transcriptional activity and significantly downregulated AR target genes at the mRNA and protein levels. Further studies proved that Z15 could bind directly to both AR LBD and AR AF1, so as to decrease androgen-induced AR nuclear translocation, which confirmed Z15 as an AR antagonist. Moreover, Z15 could also degrade AR and ARV7 through the proteasome pathway (*Figure 9*). Importantly, several Z15 analogs exhibited stronger AR inhibition and downregulation potency than Z15, suggesting that Z15 is a promising lead compound for further chemical structure optimization.

AR amplification is a common phenomenon in CRPC patients undergoing antiandrogens treatment. Our data showed that ENZa could hardly downregulate DHT-induced PSA levels in AR-overexpressed VCaP cells, which means that AR antagonizing method is too faint to overcome the drug resistance caused by AR amplification. A rational way to solve this problem would be to directly downregulate AR. For example, AR degrader ARV-110 developed based on PROTAC technology could inhibit the growth activity of AR amplification CRPC cells (*Cleveland, 2020*; *Kregel et al., 2020*). Another strategy to promote AR degradation would be to interfere with the AR protein complex stability. For example, Hu et al. discovered a p23 inhibitor named ailanthone, which could degrade AR through interfering with AR-HSP90-p23 protein complex stability (*He et al., 2016*). In addition, *Lv et al., 2021* reported a competitive AR antagonist CPPI was capable of enhancing AR interaction with its E3 ligase MDM2 and then promoting AR degradation in CRPC cells. Here, we proved that Z15 not

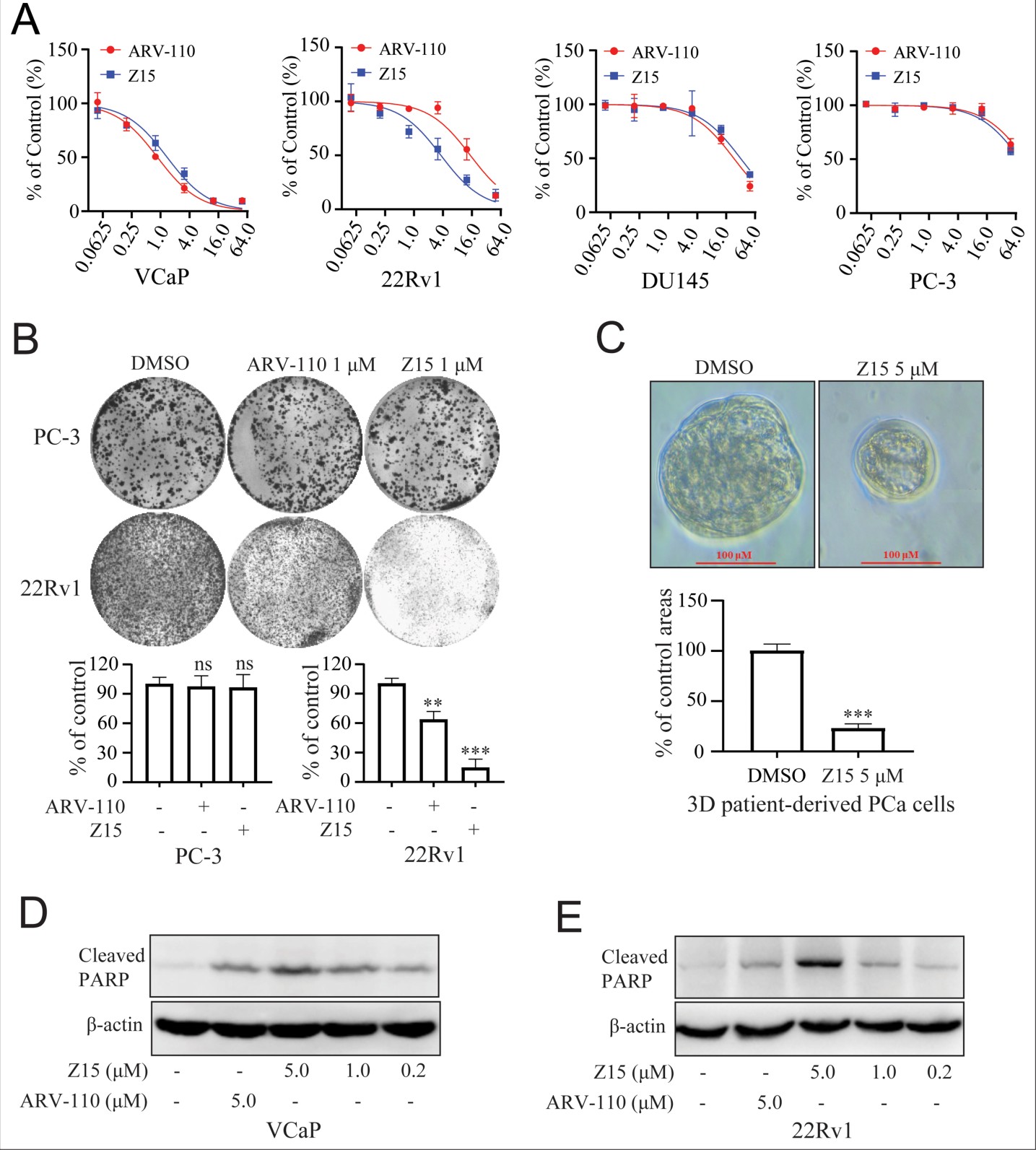

**Figure 6.** Z15 selectively inhibits proliferation and induces apoptosis of AR-positive CRPC cells. (**A**) VCaP, 22Rv1, DU145 and PC-3 cells treated with different concentrations of Z15 or ARV-110 for 72 hr, cell proliferation detected by MTT assay. (**B**) Crystal violet staining of PC-3 and 22Rv1 cells treated with or without 1 μM Z15 or ARV-110 for 12–14 days, colony numbers were quantified. (**C**) Patient-derived PCa organoid treated with 1 μM Z15 or DMSO for 7 days, then observed by microscope. (**D**) Western blot analysis of cleaved PARP protein levels in VCaP cells treated with indicated concentrations of

*Figure 6 continued on next page*

*Figure 6 continued*

Z15 for 24 hr. (**E**) Western blot analysis of cleaved PARP protein levels in 22Rv1 cells treated with indicated concentrations of Z15 for 24 hr. Experiments were performed in triplicate. Results are shown as mean ± sd. *p<0.05, **p<0.01, ***p<0.001 vs DMSO group.

The online version of this article includes the following figure supplement(s) for figure 6:

**Figure supplement 1.** Z15 promotes the apoptosis of AR-positive CRPC cell lines.

only antagonizes AR function, but also promotes AR degradation through the proteasome pathway to overcome AR amplification-induced antiandrogen resistance.

ARVs (especially ARV7) not only correlate with antiandrogen resistance but also relate to poorer prognosis of patients with CRPC (*Sharp et al., 2019*; *Antonarakis et al., 2014*). Efforts have been made to develop ARVs-targeted therapies. For instance, niclosamide has been reported to specifically downregulate ARV7 protein levels. In vitro and in vivo bioassays both indicate that niclosamide significantly inhibits the growth of CRPC driving *via* ARV7 (*Liu et al., 2014*). Unfortunately, a phase I study of niclosamide in combination with ENZa in men with CRPC reveals that oral niclosamide is not viable for repurposing as a CRPC treatment agent (*Schweizer et al., 2018*). Nevertheless, through targeting the AR N-terminal domain, some novel AR down regulators, such as UT-34, UT-69, EPI, TAS3681, and 26 f have been shown to block the transcriptional activity of both the full-length AR and ARVs. These compounds significantly inhibit the proliferation activity of ARVs-induced antiandrogen resistance CRPC cells both in vitro and in vivo (*Ponnusamy et al., 2017*; *Ponnusamy et al., 2019*; *Yang et al., 2016*; *Seki et al., 2018*; *He et al., 2021*). However, these compounds still need further clinical research. Our studies indicated that Z15 binds directly to both AR LBD and AR AF1, as a result, Z15 could inhibit AR and ARV7-driving CRPC cells such as 22Rv1 cells growth activity in vitro and in vivo. In conclusion, our data illustrate the synergistic importance of AR antagonism and degradation in advanced PCa treatment.

However, we notice that the AR degradation potency of Z15 is not as stronger as AR-targeted PROTAC molecule ARV-110, suggesting that Z15 is a less specific AR degrader compared to ARV-110. We acknowledge that while Z15 suppresses CRPC progress mainly through targeting AR and ARV7, Z15 may have other mechanisms of action which contribute to its anticancer activity in vitro and in vivo. Nevertheless, Z15 is a novel selective AR and ARV7 degrader that warrants further structure optimization and anti-CRPC mechanisms investigation.

## Methods

### Cell lines

The LNCaP, PC-3, 22Rv1, VCaP, and DU-145 human prostate cancer cell lines were purchased from American Type Culture Collection (ATCC). The identity of these cell lines has been authenticated by STR profiling, and all these cell lines tested negative for mycoplasma contamination. LNCaP (ATCC, CRL-1740) and 22Rv1 (ATCC, CRL-2505) cell lines were cultured with RPMI-1640 supplemented with 10% fetal bovine serum (FBS; PAN-Biotech, Bayern, Germany). PC-3 (ATCC, CRL-1435) was cultured with F-12K supplemented with 10% FBS (PAN-Biotech), VCaP (ATCC, CRL-2876), and DU145 (ATCC, HTB-81) cell lines were cultured with DMEM supplemented with 10% FBS (PAN-Biotech).

### Dual-luciferase reporter system assay

Plasmid wt-AR is the full-length cDNA of wild-type AR, plasmid AR F876L is the full-length cDNA of mutant AR that harbors F876L mutation, plasmid T877A is the full-length cDNA of mutant AR that harbors T877A mutation, plasmid PSA-luc is a reporter gene plasmid in which firefly luciferase expression is dependent on the PSA promoter, and plasmid Renilla is a Renilla luciferase reporter gene plasmid. The plasmids wt-AR and AR F876L were kindly provided by Dr. J.H. Wu (McGill University, Canada), plasmid PSA-luc was kindly provided by Dr. Hiroyuki (Cancer Chemotherapy Center of Japan, Japan), whereas plasmid T877A and plasmid Renilla were constructed in-house. During the reporter assays, 24 hr before transfection, PC-3 (or LNCaP, 22Rv1, VCaP) cells were seeded at a density of 6–7×10$^4$ cells per well in 24-well plates and subsequently co-transfected with 100 ng PSA-luc, 20 ng wt-AR (or 20 ng AR F876L, or 20 ng AR T877A, or their absence in LNCaP, 22Rv1, and VCaP cells), and 3 ng Renilla plasmids using Lipofectamine 2000 reagent (Invitrogen, Waltham, MA, USA)

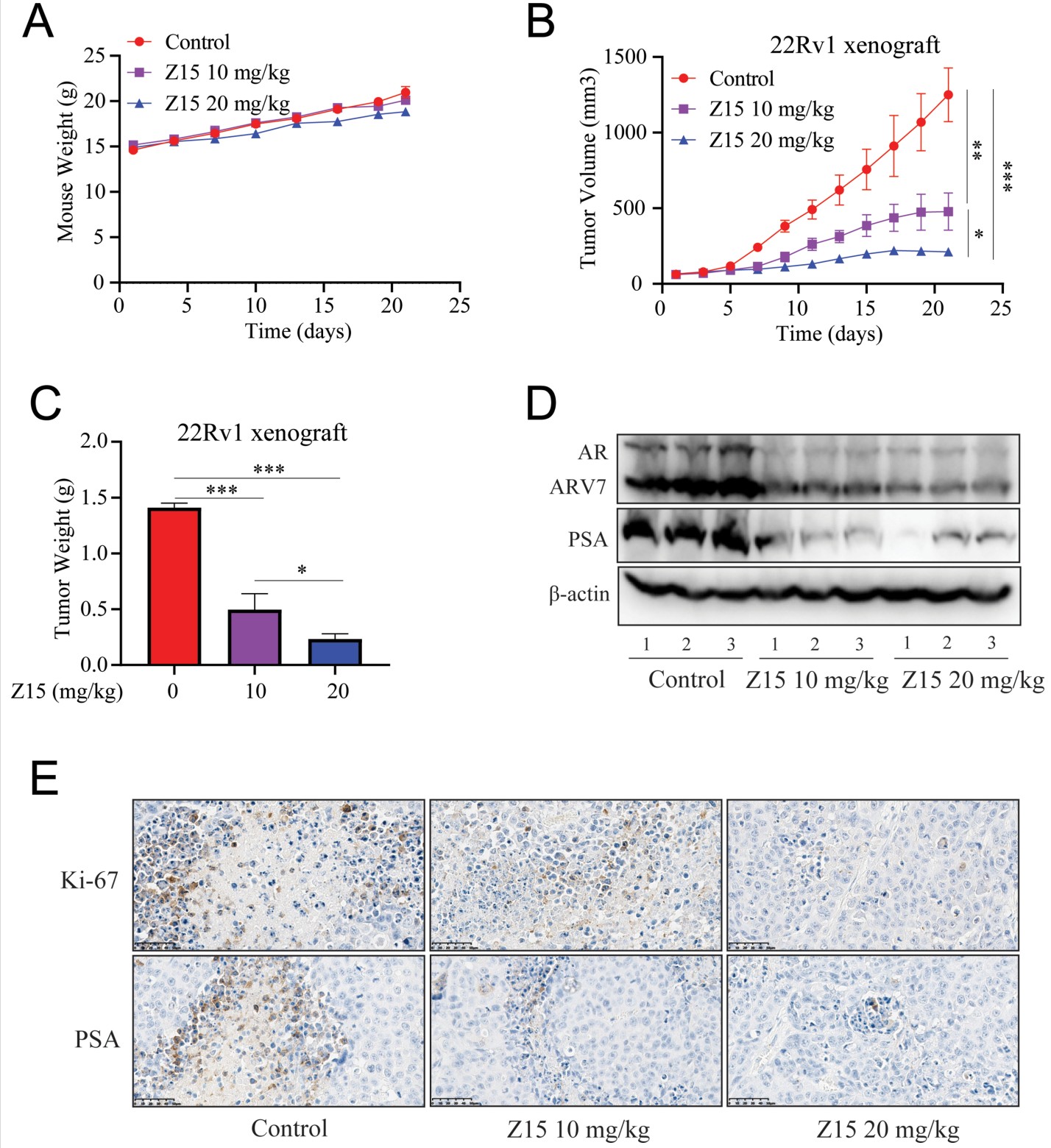

**Figure 7.** Z15 suppresses 22Rv1 xenografts progress in vivo. (**A**) Xenografts arising from 22Rv1 cells treated with blank control, 10, or 20 mg/kg Z15 once a day for 21 days. Mice weighed by electronic scale. (**B**) Tumor growth monitored every other day. (**C**) Tumor weight monitored on the last day. (**D**) Western blot analysis of AR, ARV7, and PSA protein levels in lysed tumor tissues. (**E**) Immunohistochemical analysis of proliferation (Ki67) and PSA levels in harvested tumors. Scale bar represents 50 μm. Results are shown as mean ± sd. *p<0.05, **p<0.01, ***p<0.001 vs control group.

*Figure 7 continued on next page*

Figure 7 continued

The online version of this article includes the following figure supplement(s) for figure 7:

**Figure supplement 1.** Z15 reduced AR, ARV7 and PSA levels in vivo.

following the manufacturer's protocol. Then, 24 hr after the transfection, the medium was changed to phenol red-free RPMI-1640 supplemented with 10% charcoal-stripped FBS, containing 5 nM DHT (1 µL) and 1 µL test compounds at the designated concentrations. After a further 24 hr incubation, the cells were lysed in 100 µL passive lysis buffer per well, and 20 µL cell lysates were used for detection of the luciferase activity using a Dual-luciferase Assay System (Promega, Madison, WI, USA) on a plate reader (Centro XS3 LB 960; Berthold, Berlin, Germany). All experiments were run in triplicate.

## Western blot

Cells were seeded at a density of $3–4×10^5$ cells per well in six-well plates. After 48 hr incubation, 4 µL DMSO (Sigma-Aldrich, St. Louis, MO, USA) or 4 µL test compounds were added to each well at the designated concentrations. After another 24 hr incubation, the cells were lysed with 60 µL radio-immunoprecipitation assay lysis buffer. Then the protein lysis buffer was treated with 10% SDS-PAGE and transferred onto polyvinylidene difluoride membranes for western blot analysis. The following antibodies were used for the detection of proteins: rabbit anti-AR (#sc-816, 1:1000; Santa Cruz, SANTA CRUZ, CA, USA), mouse anti-AR (#sc-7305, 1:1000; Santa Cruz), mouse anti-PSA (#sc-7316, 1:1000, Santa Cruz), rabbit anti-cleaved PARP (#5625T, 1:3,000; Cell Signaling Technology, Danvers, MA, USA), rabbit anti-GR (#12041 S, 1:1000, Cell Signaling Technology), mouse anti-HSP90 (#ab13492, 1:1000, Abcam, Cambridge, UK), mouse anti-CDK7 (#sc-7344, 1:1000, Santa Cruz). Mouse anti-β-actin (#ab8226, 1:5000; Abcam) was used as a loading control. Proteins were visualized using anti-mouse, anti-rabbit, or anti-goat HRP-conjugated secondary antibodies (1:5000; Zhongshan Jinqiao Biotechnology, Shanghai, China) and ECL-Plus (Millipore, Burlington, MA, USA). The resulting bands were analyzed and quantified using ImageJ v1.49g (National Institutes of Health, Bethesda, MD, USA). Each experiment was repeated at least twice.

## MTT assay

Cells were seeded at a density of $1×10^4$ cells per well in 96-well plates. VCaP and 22Rv1 cells were cultured in a complete RPMI-1640 growth medium, PC-3 cells were cultured in an F-12K growth medium, and DU-145 cells were cultured in a DMEM growth medium. After 24 hr incubation, 1 µL DMSO (Sigma-Aldrich) or 1 µL test compounds were added to each well at the designated concentrations. After 72 hr incubation, 20 µL MTT (Invitrogen) solution (5 mg/mL in PBS) was added per well and incubated for another 4 hr. The MTT formazan formed by metabolically viable cells was dissolved in 100 µL isopropanol. The absorbance was measured at 570 nm wavelength on a plate reader (EnSpire 2300; PerkinElmer, Waltham, MA, USA). Experiments were performed in triplicate. The value of the DMSO group was defined as 100%.

## Colony formation assay

PC-3 and 22Rv1 cells were seeded at a density of $4×10^2$ cells per well in six-well plates. After 24 hr incubation, 4 µL DMSO (Sigma-Aldrich) or 4 µL Z15 and ARV-110 were added to each well at the designated concentration. The first culture media change occurred after another 48 hr incubation and then repeated every three days for 2 weeks until cell colony formation. Then, the culture media was removed and the cells were washed twice with PBS, 1 mL methanol was added to each well for 10 min, the cells were treated with 1 ml Giemsa stain for 10 min, and each well was washed with water until the background was clean, and then aired for another 30 min before taking pictures of the colonies. To quantify staining, the stained wells were washed with 1 mL 10% acetic acid, and absorbance was detected at a wavelength of 590 nm on a plate reader (EnSpire 2300; PerkinElmer), The value of the DMSO group was used as control and defined as 100%. Experiments were performed in triplicate.

## PCa organoid isolation and culture

Digest the PCa tissues in 5 mg/ml Collagenase II with 10 µM Y-27632 in a 15 ml Falcon tube at 37 °C on a shaking platform for 4 h. Use 1 ml of 5 mg/ml Collagenase II per ~50 mg minced tissue.

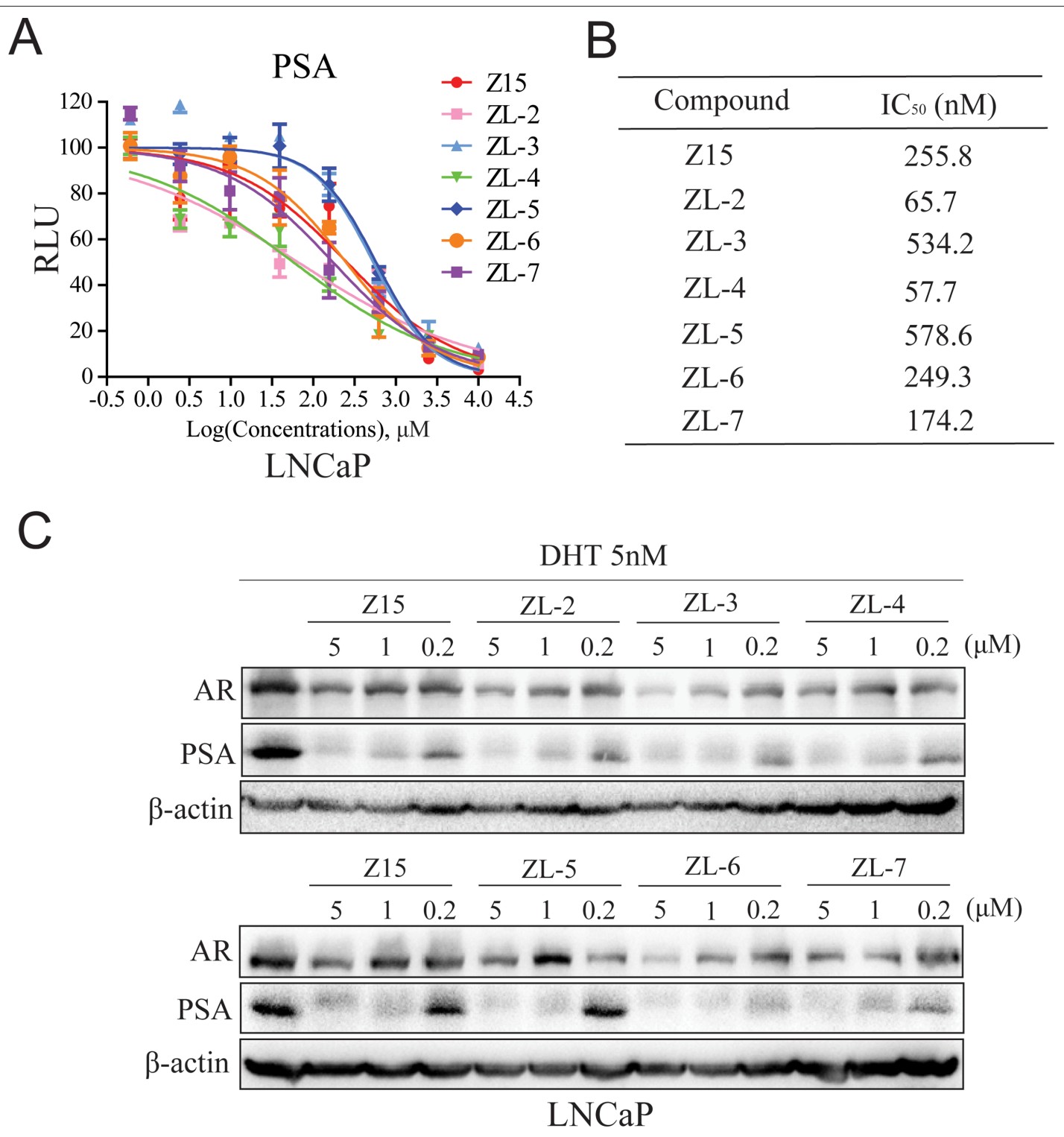

**Figure 8.** Z15 analogs show comparable AR inhibition activity. (**A**) Dual-luciferase reporter assay to measure PSA-luc reporter luciferase activities in LNCaP cells stimulated by 5 nM DHT, and treated with different concentrations of indicated compounds for 24 hr. (**B**) AR transcription inhibition IC$_{50}$. (**C**) Western blot analysis of PSA and AR protein levels of lysed LNCaP cells treated with indicated concentrations of Z15 and its analogs in the presence of 5 nM DHT for 24 hr. Results are shown as mean ± sd. Experiments were performed in triplicate.

The online version of this article includes the following figure supplement(s) for figure 8:

**Figure supplement 1.** The chemical structure of seven Z15 analogues.

*Figure 8 continued on next page*

**Figure supplement 2.** Z15 analogues could inhibit AR transcription activity and reduce AR protein levels.

**Figure supplement 3.** Z15 analogs downregulate AR and PSA protein levels.

The tumor tissue fragments were digested into epithelial cell clusters, washed with ice PBS and centrifuged for 1 min at 300×g. Then the precipitate was suspended in the precooled matrigel. The suspension was inoculated in a 24-well plate and incubated in a 37 °C incubator for 30 min until the matrigel became gelatinous. Then, 500 µL medium per well was added, the PCa organoid medium was prepared as follows: Add 1.0 ml B27, 500 µl nicotinamide (1 M in PBS), 125.0 µl N-acetylcysteine (500 mM in PBS), 0.5 µl of EGF (0.5 mg/ml in PBS + 0.1% BSA), 5.0 µl A83-01 (5 mM in DMSO), 50.0 µl Noggin (100 µg/ml in PBS + 0.1% BSA), 50.0 µl R-spondin 1 (500 µg/ml in PBS + 0.1% BSA or 10% conditioned medium), 50.0 µl dihydrotestosterone (1 µM in ethanol), 5.0 µl FGF2 (50 µg/ml in PBS + 0.1% BSA), 5.0 µl FGF10 (0.1 mg/ml in PBS + 0.1% BSA), 5.0 µl prostaglandin E2 (10 mM in DMSO), 16.7 µl SB202190 (30 mM in DMSO) and top up to 50 ml with adDMEM/F12 (containing penicillin/streptomycin, 10 mM Hepes and GlutaMAX 100×diluted). After passaging, Y-27632 is added to the culture medium (e.g. add 5.0 µl of 100 mM to 50 ml human prostate culture medium). The culture medium was refreshed every 3 days, 7 days after PCa organoid cultured, add 1 µL DMSO or Z15 at the designated concentration, then cultured for another 7 days.

### Quantitative real-time PCR

Cells were seeded at a density of $3–4×10^5$ cells per well in six-well plates and cultured with RPMI-1640 with 10% charcoal-stripped FBS for 48 hr before treatment with 4 µL DMSO (Sigma-Aldrich) or 4 µL test compounds added to each well at the designated concentrations. After another 24 hr incubation, the total RNA was extracted using TRIzol reagent (Invitrogen) according to the manufacturer's instructions. The total RNA (1 µg) was used for complementary DNA synthesis using a cDNA reverse transcription kit (Takara, Shiga, Japan). Real-time PCR was performed in triplicate using gene-specific primers on the PCR instrument (Stratagene MX3000P; Agilent, Santa Clara, CA, USA). The mRNA levels were normalized to GAPDH. The gene-specific primers are listed in *Supplementary file 1b*.

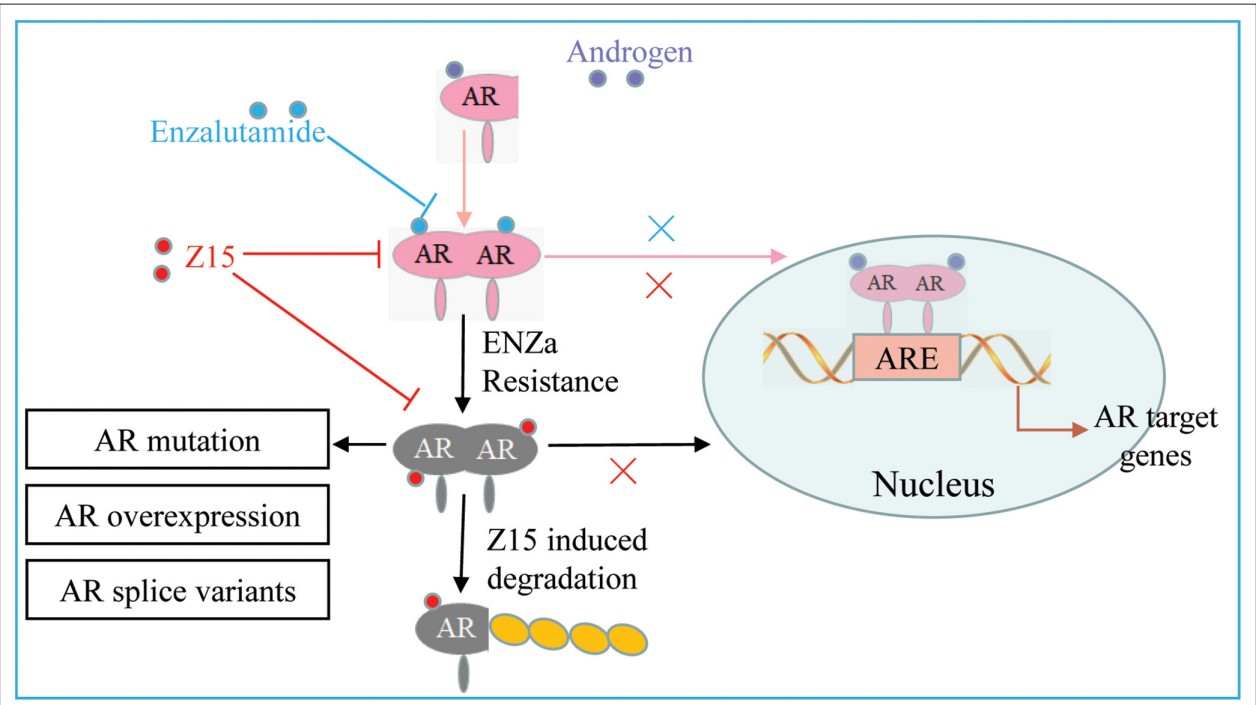

**Figure 9.** The mechanism of Z15 inhibits the AR pathway and overcomes antiandrogen resistance. Z15 binds to both AR LBD and AR AF1, decreases AR nuclear translocation, antagonizes AR function, promotes AR and ARVs degradation through the proteasome pathway, so as to overcome AR mutation, AR overexpression, and ARVs-induced antiandrogen resistance.

## Bio-layer interferometry assay

The bio-layer interferometry measurements were conducted using a ForteBio Octet RED96 instrument (ForteBio, Fremont, CA, USA) equipped with super streptavidin biosensor chips (ForteBio). For the AR LBD binding analysis, purified AR protein (A15675; Thermo Fisher Scientific, Waltham, MA, USA) was biotinylated *via* EZLink NHS-Biotin Reagents (#21343; Thermo Fisher Scientific) at a ratio of 1:1 for 1 hr at 25 °C. A duplicate set of Super streptavidin-coated biosensors were incubated in PBS, then moved to wells containing 0.02% Tween-20 PBS solution with 50 μg/mL biotinylated AR protein for 10 min. Non-specifically bound AR was removed with a 1 min wash step, then the baseline was detected at the same wells for 1 min, and AR-coated biosensors were moved to wells containing 250 μM Z15 and 0.02% Tween-20 PBS solution for the association. The biosensor was then moved back to the baseline solution and dissociation of Z15 from the biosensors was detected. The baseline-association-dissociation steps were repeated for the adjacent wells containing 125, 62.5, 31.25, and 15.6 μM Z15, respectively. A negative control of 1% DMSO was also used during these steps. Association and dissociation of samples and the negative control to coated biotinylated AR sensors were measured for 1 min. Data analysis on the ForteBio Octet RED instrument was performed using a reference well subtraction in the ForteBio data analysis software. Binding between ENZa and AR was performed by the same method.

For the AR AF1 binding analysis, the AR-AF1 domain (110–486 aa) was cloned into the Pet21a vector and transformed into BL21-competent cells. Expression of the protein was induced with 0.5 mM IPTG when the optical densitometry reached 0.6 and then incubated at 37 °C for 5 hr. Cells were harvested and lysed in a lysis buffer (10 mM HEPES pH 7.4, 100 mM NaCl, 10% glycerol, and protein inhibitors). The AR-AF1 protein was first purified with affinity chromatography (Ni Sepharose excel) using the ÄKTA chromatography system (Cytiva Life Sciences, Marlborough, MA, USA). Peak fractions were pooled and further purified with anion exchange (Q Sepharose Fast Flow) chromatography. Binding affinities between AR AF1 and ENZa, Z15, and UT-34 were performed by the same method as described above.

## Competitive AR binding assay

LNCaP cells were seeded at a density of $2 \times 10^5$ cells per well in 12-well plates cultured with RPMI-1640 media supplemented with 10% FBS. After 48 hr incubation, RPMI-1640 media was removed and cells were washed with PBS, then cultured with phenol-free RPMI-1640 media supplemented with 2% BSA mixed with 1 nM [$^3$H] DHT (#2645285; PerkinElmer) and 5 μL test compounds at the designated concentrations. Thereafter, 5 μL 10 μM R1881 was transferred to the assay plate for complete competitive binding (background) and DMSO was used for total binding (hole coverage). After incubating at 37 °C for 4 hr, the cells were washed twice with ice-cold Tris-buffered saline and then lysed with a solution of 200 μL 0.2 M NaOH. Isolated cell samples were counted using a scintillation counter, with appropriate standards of total activity and blank controls. The 'Inhibition [% Control]' was calculated using the equation: %Inhibition = (1 − background-subtracted assay value/background-subtracted hole coverage value)×100. Experiments were performed in triplicate.

## Surface Plasmon Resonance assay

AF1 was cross coupled with EDC and NHS on CM5 sensor chip to obtain ~7000 RU AF1 immobilized on chip surface. The experiment was conducted on method model, and performed at 30 μL/min, 25 °C. Three buffer blanks were added to the beginning, followed by the compound samples. Association and dissociation time were set to 120 s. The needle was extra washed with 50% DMSO in every cycle, DMSO effects between channels was normalized with 1% DMSO controls.

Sensorgrams for all binding interactions were recorded in real time and analyzed after subtracting that from reference channel and internal blanks. Then, sensorgrams were fitted with kinetics or steady-state affinity model. A set of predefined models for kinetic and steady-state affinity are provided with Biacore 8 K Evaluation Software. KD was fit by Biacore 8 K Evaluation Software.

## Immunofluorescence

LNCaP cells were grown on the glass cell culture dish in phenol red-free RPMI-1640 medium containing 10% charcoal-stripped FBS for 24 h, and then treated with either DMSO, 5 nM DHT, 5 μM ENZa with 5 nM DHT, or 5 μM Z15 with 5 nM DHT, respectively for further 4 hr. Cells were fixed with 4% (vol/vol)

paraformaldehyde and permeabilized with 0.2% Triton X-100. Then the cells were incubated with AR antibody (N-20, 1:100; Santa Cruz). The secondary antibody, goat anti-rabbit with FITC (Santa Cruz) was used at a ratio of 1:100. The counterstain DAPI was used to visualize the cell nucleus. The images were detected under an UltraVIEW vox spinning disc confocal scanning system (Perkin Elmer) on an Olympus IX81 microscope.

## In vivo xenograft study

BALB/c nude male mice (5 weeks old from the Institute of Experimental Animals, Chinese Academy of Medical Sciences, Beijing, China) were subcutaneously injected with $1\times10^7$ 22Rv1 cells. When the average tumor volume reached 50–100 $mm^3$, the mice were castrated under Nembutal anesthesia and then randomized into three groups (n=3). Oral treatments with Z15 (10 or 20 mg/kg) or vehicle were administered daily and continued for 3 weeks. The tumor volume was monitored every other day, the body weight was monitored every three days, and the tumor volume was calculated according to the formula $W 2 \times L/2$ ($mm^3$), where $W$ was the short diameter and $L$ was the long diameter. All animal experiments have been approved by the Ethics Committee of Nanjing Lambda Pharmaceutical Co., Ltd (Reference number: IACUC-20210902).

## Immunohistochemical staining

Formalin-fixed paraffin-embedded sections were stained with hematoxylin and eosin. IHC staining was performed with antibodies against Ki67 (ab16667; Abcam) and PSA (cy5775; Abways, Shanghai, China). Generally, the rehydrated slides were microwave-heated for 20 min in citrate buffer (10 mM, pH 6.0) for antigen retrieval. Then, the slides were incubated in 1% $H_2O_2$ for 10 min, blocked with serum-free protein block, and incubated with the primary antibodies (Ki67, 1:200; PSA, 1:50) for 2 hr at 25 °C, followed by incubation with HPR-conjugated secondary antibody for 1 hr at 25 °C. The immunoreaction products were visualized with 3, 3-diaminobenzidine/$H_2O_2$ solution.

## Immunoprecipitation

22Rv1 cells co-transfected with pRBG4-CW7-myc-ubiquitin (cDNA encoding Myc-tagged human ubiquitin, kindly provided by Xiaofang Yu) were treated with or without 5 µM Z15 in the presence of 5 µM MG132 for 12 hr. Then, the cells were washed with ice-cold PBS and collected in NP-40 buffer. The lysates were lysed for 1 hr in an ice bath, then centrifuged at 12,000 $g$ for 10 min. The supernatants were incubated with 5 µL antibody to AR (sc-7305; Santa Cruz) overnight at 4 °C, with 40 µL protein A/G and rocked for 4 hr at 4 °C. The protein A/G beads were pelleted and washed twice with ice-cold PBS. The precipitates were resolved on an SDS-polyacrylamide gel electrophoresis gel and subjected to Western blot analysis.

## 4D-label free proteomic analysis

LNCaP Cells were seeded at a density of $1.5\times10^6$ cells in T25 culture flask. After 48 hr incubation, 5 µL test compounds were added to each flask at the designated concentrations. After another 8 hr incubation, the cell samples were lysed and the proteins were digested according to the manufacturer's instructions.

The tryptic peptides were dissolved in solvent A (0.1% formic acid, 2% acetonitrile/in water), and directly loaded onto a homemade reversed-phase analytical column (25 cm length, 75/100 µm i.d.). Peptides were separated with a gradient from 6% to 24% solvent B (0.1% formic acid in acetonitrile) over 70 min, 24% to 35% in 14 min and climbing to 80% in 3 min then holding at 80% for the last 3 min, all at a constant flow rate of 450 nL/min on a nanoElute UHPLC system (Bruker Daltonics).

The peptides were subjected to capillary source followed by the timsTOF Pro (Bruker Daltonics) mass spectrometry. The electrospray voltage applied was 1.60 kV. Precursors and fragments were analyzed at the TOF detector, with a MS/MS scan range from 100 to 1700 m/z. The timsTOF Pro was operated in parallel accumulation serial fragmentation (PASEF) mode. Precursors with charge states 0–5 were selected for fragmentation, and 10 PASEF-MS/MS scans were acquired per cycle. The dynamic exclusion was set to 30 s.

The resulting MS/MS data were processed using MaxQuant search engine (v.1.6.15.0). Tandem mass spectra were searched against the human SwissProt database (20422 entries) concatenated with reverse decoy database. Trypsin/P was specified as cleavage enzyme allowing up to 2 missing

cleavages. The mass tolerance for precursor ions was set as 20 ppm in first search and 5 ppm in main search, and the mass tolerance for fragment ions was set as 0.02 Da. Carbamidomethyl on Cys was specified as fixed modification, and acetylation on protein N-terminal and oxidation on Met were specified as variable modifications. FDR was adjusted to <1%. The cut-off values for identification of potential hits were set at Fold Change <0.67 or Fold Change >1.5 and p-value <0.05.

Enrichment of pathway analysis: Encyclopedia of Genes and Genomes (KEGG) database was used to identify enriched pathways by a two-tailed Fisher's exact test to test the enrichment of the differentially expressed protein against all identified proteins. The pathway with a corrected p-value <0.05 was considered significant. These pathways were classified into hierarchical categories according to the KEGG website. The mass spectrometry proteomics data have been deposited to the ProteomeXchange Consortium via the PRIDE (*Perez-Riverol et al., 2022*) partner repository with the dataset identifier PXD035721.

## In silico screening

The AR antagonist common molecular characteristic pharmacophore model and the antagonistic form AR-LBD were constructed as previously described (*Wu et al., 2019*). About 7.5 million compounds were downloaded from the ZINC drug-like database and ChemDiv database. Then, these 2D molecules were generated into 3D structures using the 'prepare ligands' module by Discovery Studio v3.5. Next, pharmacophore-based virtual screening was performed by Discovery Studio v3.5. Molecules with a Fit-value >4 were chosen for further screening. The left molecules were then docked into the HBP of the antagonistic form AR-LBD using the GOLD software and ranked by fitness score. The absorption, distribution, metabolism, excretion, and toxicity (ADMET) of compounds were evaluated by the 'General Descriptors' module in Discovery Studio v3.5.

## Chemistry

### 2-romo-1-(2-thienyl)-ethanone (2)

Add a solution of 2-Acetylthiophene (10 mmol) in CH2Cl2 (4 mL) dropwise to a solution of N-bromosuccinimide (12 mmol, 1.2 equiv) and *p*-TsOH (1 mmol, 0.1 equiv) in DCM (10 mL) at 0 °C. Heat the reaction mixture at reflux for 4 hr. The reaction was monitored by TLC. After completion of the reaction, it was quenched by water (20 mL) and extracted with ethyl acetate (3×20 mL). The organic layer was dried over $Na_2SO_4$ and concentrated to give a residue. The residue was purified by a silica gel column eluted with pet ether and ethyl acetate to give the yellow liquid 2. Yield 63%.

### 4-(2-Thienyl)-2-thiazolamine (4)

Add 2-bromo-1-(2-thienyl)-ethanone (5 mmol) to a solution of thiourea (6 mmol, 1.2 equiv) in EtOH (5 mL) at RT. Heat the reaction mixture at reflux for 2 hr. Reaction mixtures changed from faint yellow to orange during the course of the reaction, with the formation of a precipitate. The mixture was then allowed to cool to RT. After reaching RT, the mixture was filtered with a Buchner funnel. The precipitate was then rinsed with ethyl acetate, in order to remove excess thiourea, which generated a yellow solid. Yield 90%.

### 4-(Bromomethyl)-3,5-dimethyl-1,2-oxazole (6)

To a solution of 3,5-Dimethyl-4-hydroxymethylisoxazole (10 mmol) in anhydrous DCM (10 mL) was added dropwise phosphorus tribromide (20 mmol, 2.0 equiv) at 0 °C. Heat the reaction mixture at RT for 3 hr. The reaction was monitored by TLC. After completion of the reaction, it was quenched by Saturated aqueous sodium bicarbonate solution and extracted with ethyl acetate (3×20 mL). The organic layer was dried over $Na_2SO_4$ and concentrated to give a residue. The residue was purified by a silica gel column eluted with pet ether and ethyl acetate to give the pellucid liquid 6. Yield 92%.

### 4-[(3,5-Dimethyl-4-isoxazolyl) methoxy]-benzoic acid (8)

Add KOH (12.5 mmol, 2.5equiv) and 4-(bromomethyl)–3,5-dimethyl-1,2-oxazole (5 mmol, 1.0equiv) to a solution of 4-hydroxybenzoic acid (5 mmol, 1equiv) in EtOH:$H_2O$ 9:1 (5 ml). Heat the mixture under reflux for 12 hr. The mixture was then allowed to cool to RT. The mixture was acidified by 20% HCl

solution with white precipitate formed. The mixture was filtered with a Buchner funnel and rinsed with 95% EtOH, which generated a white solid 8. Yield 60%.

### 4-[(3,5-Dimethyl-4-isoxazolyl) methoxy]-N-4-[(2-thienyl)-2-thiazole]-benzamide (Z15)

Add 4-(2-thienyl)–2-thiazolamine (3 mmol), 1.0equiv to a solution of 4-[(3,5-dimethyl-4-isoxazolyl) methoxy]-benzoic acid (3 mmol, 1.0 equiv), EDCI (4.5 mmol, 1.5 equiv), HOBT (3.6 mmol, 1.2 equiv), DMAP (0.36 mmol, 0.12 equiv) and $Et_3N$ (9 mmol, 3.0 equiv) in DCE (30 mL) at RT. Heat the reaction mixture at reflux for 24 hr. The reaction was monitored by TLC. After completion of the reaction, it was then allowed to cool to RT and quenched by Saturated brine (20 mL). The mixture was extracted with DCM (3×20 mL). The organic layer was dried over $Na_2SO_4$ and concentrated to give a residue. The residue was purified by a silica gel column eluted with pet ether and ethyl acetate to give the white solid Z15. Yield 66%.

Z15. White solid (813.8 mg), [1]H NMR (600 MHz, DMSO) δ 12.71 (s, 1 H), 8.19 (d, $J$=8.7 Hz, 2 H), 7.59 (d, $J$=3.0 Hz, 1 H), 7.56 (s, 1 H), 7.54 (d, $J$=4.9 Hz, 1 H), 7.20 (d, $J$=8.7 Hz, 2 H), 7.17–7.13 (m, 1 H), 5.07 (s, 2 H), 2.47 (s, 3 H), 2.27 (s, 3 H). HR MS (ESI), m/z: 412.0789 [M+H]+.

### Data analysis

All statistical analyses were performed using Prism v5.01 (GraphPad Software, San Diego, CA, USA). Except where specified, comparisons between the groups were performed using a two-tailed Student's $t$-test, and the differences were considered statistically significant for $P<0.05$.

## Acknowledgements

We thank Dr. Jianhui Wu (Mcgill University), and Dr. Hiroyuki Seimilya (Cancer Chemotherapy Center, Japan) for providing AR-expressing plasmids. We thank the equipment support from Public Laboratory Platform, National Science and Technology Key Infrastructure on Translational Medicine in Peking Union Medical College Hospital. This work was in part supported by the Nature Science Foundation of China (22077115, 82104231, 81672559, 81311120299) and China Postdoctoral Science Foundation (2021M700504).

## Additional information

### Funding

| Funder | Grant reference number | Author |
| --- | --- | --- |
| National Natural Science Foundation of China | 22077115 | Jinming Zhou |
| National Natural Science Foundation of China | 82104231 | Meng Wu |
| China Postdoctoral Science Foundation | 2021M700504 | Meng Wu |
| National Natural Science Foundation of China | 81672559 | Jinming Zhou |
| National Natural Science Foundation of China | 81311120299 | Jinming Zhou |

The funders had no role in study design, data collection and interpretation, or the decision to submit the work for publication.

### Author contributions

Meng Wu, Data curation, Methodology, Writing - original draft; Rongyu Zhang, Data curation, Validation, Methodology; Zixiong Zhang, Data curation, Methodology; Ning Zhang, Chenfan Li, Fangjiao Huang, Ruoying Zhang, Methodology; Yongli Xie, Validation, Methodology; Haoran Xia, Data curation; Ming Liu, Resources, Methodology; Xiaoyu Li, Data curation, Formal analysis, Validation; Shan

Cen, Supervision, Validation, Project administration; Jinming Zhou, Conceptualization, Supervision, Funding acquisition, Validation, Investigation, Project administration, Writing - review and editing

## Author ORCIDs
Meng Wu ⓘ http://orcid.org/0000-0001-7381-5591
Rongyu Zhang ⓘ http://orcid.org/0000-0001-7361-9152
Jinming Zhou ⓘ http://orcid.org/0000-0003-1610-5061

## Ethics
All animal experiments have been approved by the Ethics Committee of Nanjing Lambda Pharmaceutical Co.,Ltd (Reference number: IACUC-20210902).

## Decision letter and Author response
Decision letter https://doi.org/10.7554/eLife.70700.sa1
Author response https://doi.org/10.7554/eLife.70700.sa2

# Additional files

## Supplementary files
• Supplementary file 1. Supplementary tables described in this article. (a) The chemical structures of 80 candidate compounds. (b) Primer sequences for qRT-PCR. (c) The spectroscopic data of the Z15 derivatives. (d) Decreased protein profiles of LNCaP cells treated with ARV-110 plus DHT compared to DHT detected by 4D-lael free proteomics. (e) Decreased protein profiles of LNCaP cells treated with Z15 plus DHT compared to DHT detected by 4D-lael free proteomics

• Transparent reporting form

• Source data 1. Original gel and blot.

• Source data 2. Figure supplement source data.

• Source data 3. *Figures 1–8* source data.

## Data availability
All data generated or analysed during this study are included in the manuscript and supporting source files. The RNA sequence data could be found in the following link: https://bigd.big.ac.cn/gsa-human/browse/HRA000921. The mass spectrometry proteomics data have been deposited to the ProteomeXchange Consortium via the PRIDE partner repository with the dataset identifier PXD035721. PRIDE - Proteomics Identification Database.

The following datasets were generated:

| Author(s) | Year | Dataset title | Dataset URL | Database and Identifier |
|---|---|---|---|---|
| Zhou J | 2023 | Selective androgen receptor degrader (SARD) to overcome antiandrogen resistance in castration-resistant prostate cancer | https://www.ebi.ac.uk/pride/archive/projects/PXD035721 | PRIDE, PXD035721 |
| Zhou J | 2021 | Selective androgen receptor degrader (SARD) to overcome antiandrogen resistance in castration-resistant prostate cancer | https://ngdc.cncb.ac.cn/gsa-human/browse/HRA000921 | Genome Sequence Archive, HRA000921 |

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
