## [Editor Report]

The present study reports the discovery and preclinical evaluation of a novel therapeutic agent for the treatment of castration-resistance prostate cancer through inducing degradation of androgen receptor. The major strength of this study is the identification of a novel lead compound and its interesting in vitro and in vivo activities in prostate cancer models.

---

## [Decision Letter]

**Decision letter after peer review:**

Thank you for submitting your article "Selective androgen receptor degrader (SARD) to overcome antiandrogen resistance in castration-resistant prostate cancer" for consideration by *eLife*. Your article has been reviewed by 3 peer reviewers, and the evaluation has been overseen by a Reviewing Editor and Wafik El-Deiry as the Senior Editor. The following individuals involved in review of your submission have agreed to reveal their identity: Frank Cackowski (Reviewer #1).

Essential revisions:

The data has novel and interesting findings with the small molecule for potential further development in castrate-resistant prostate cancer but the reviewers cautioned against over-interpreting or over-selling the proposed mechanism. The manuscript requires extensive proofreading for english language.

*Reviewer #1 (Recommendations for the authors):*

Despite having the ability to grow in the presence of low concentrations of androgens, most cases of castration resistant prostate cancer retain activity of signaling through the androgen receptor (AR) and employ one or more of several resistance mechanisms including amplification of the AR gene, point mutations in the AR gene, or splice variants such as ARv7 which lack the AR ligand binding domain and signal in a ligand independent fashion. Therefore, a key therapeutic approach in the field is to develop methods of inhibiting these mutant or amplified forms of AR. A particularly active area of work is the development of selective androgen receptor degraders, which not only inhibit AR, but cause its destruction. Some molecules are in clinical trials, but more are needed. Here the authors discovered a new molecule, named Z15, which they propose binds to the AR ligand binding domain and causes AR destruction by way of the proteasome; therefore acting as a selective androgen receptor degrader (SARD). ;

Strengths: The discovery of Z15 and its efficacy are quite promising:

1. Z15 is effective against AR expressing cell lines in vitro including cell lines expressing mutant forms of AR. In support of the specificity of the molecule, it has minimal activity against prostate cancer cell lines which do not express AR.

2. Z15 is active in a mouse xenograft prostate cancer model and is tolerable to the mice, as it does not cause weight loss.

3. IC50 and Kd values are similar to the commonly used anti-androgen, enzalutamide.

Weaknesses: The proposed mechanism of action of Z15 is not supported by the data. Therefore, Z15 cannot accurately be called a selective androgen receptor degrader. However, Z15 does appear to act through the androgen receptor in some, as of yet, unknown fashion.

1. Despite proposing that Z15 binds to the AR ligand binding domain, it appears to have similar efficacy against wild type AR and the splice isoform ARv7 which lack the AR ligand binding domain. For example, please see figure 2E. Z15 has similar effects on wild type AR (top band) and ARv7 (bottom band). Multiple high profile publications have reported lack of the ligand binding domain in ARv7. For example, see Antonarakis et al. N Engl J Med 2014; 371:1028-1038.

2. The data that Z15 affects AR protein stability as opposed to AR protein expression by some other means is not convincing. For example, see the cyclohexamide experiment in figure 5D. There is minimal difference between the cyclohexamide minus conditions (left half of figure) and the cyclohexamide containing conditions (right half of figure).

I think Z15 is a useful molecule, but please do not over-sell the data. I do not think that it acts through your proposed mechanism. However, Z15 is specific to AR expressing cell lines, and efficacious in vivo. I suggest only publishing that much of the story and leaving the downstream mechanism to a subsequent publication.

The manuscript is in need of English language editing.

*Reviewer #2 (Recommendations for the authors):*

The manuscript reports the discovery of a novel selective androgen receptor degrader (SARD) targeting the androgen receptor ligand binding domain (AR-LBD) via virtual screening and bioassays. However, the potential targets and binding sites for the degradation of AR and ARVs have not been clarified because the binding of AR-LBD may not induce the degradation of ARVs. For instance, Ponnusamy et al. (10.1158/0008-5472.CAN-17-0976) reported SARD not only binds to AR LBD but also binds to the amino-terminal transcriptional activation domain (AF-1) of AR. Overall, this paper is not suitable to be published on this journal.

1. The authors demonstrate that Z15 is a SARD with glucocorticoid receptor (GR) transcription inhibition activity. The authors should test the transcription inhibition activities of Z15 against progesterone receptor (PR) and mineralocorticoid receptor (MR) due to their high structural similarities.

2. Lines 110-111. The authors state 'The results indicated that Z15 could inhibit DHT-induced transcriptional activities of both exogenous and endogenous AR in a dose-dependent manner (Figure 1B, 1C, 1D).' However, it seems Z15 inhibits VCaP not in a dose-dependent manner (Figure 1D).

3. The authors describe and interpret their data using the word 'significantly' for the Western blot result. Quantitative results for all Western blot results in the manuscript should be presented, such as using the densitometric analysis of the band intensities.

4. The authors should quantitatively analyze the AR nuclear localization results in Figure 3.

5. Positive control should be added in Figure 6A-F, such as Enzalutamide.

6. The loading controls (β-action) are uneven, indicating the samples are not quantified before Western blot analysis

*Reviewer #3 (Recommendations for the authors):*

The present study reports the discovery and preclinical evaluation of a novel therapeutic agent for the treatment of castration-resistance prostate cancer through inducing degradation of androgen receptor.

The major strengths of this study is the identification of a novel lead compound and its interesting in vitro and in vivo activities in prostate cancer models. The major weaknesses with this study are that since the concept of induced AR degradation is not new, additional data are needed to demonstrate some major advantages of the agent reported by the authors over current AR degraders and its potential for clinical translation.

The authors have identified a good lead compound but further optimization and studies are needed to make a true impact for the filed of developing AR degraders for the treatment of castration-resistance prostate cancer.

This study reported identification of a novel selective androgen receptor degrader molecule (SERD), which the authors named Z15. The authors have provided data to demonstrate the following for Z15:

1) inducing degradation of wild-type AR and AR-V7 in a proteasome-dependent manner;

2) displaying fairly potent cell growth inhibition of AR+ LNCaP, VCaP and 22RV1 cell lines and some selectivity over AR- PC-3 and DU-145 cell line;

3) binding to AR ligand binding domain (LBD) directly with a Kd value of 21 μm and blocking the binding of AR agonist with AR LBD;

4) inhibiting 22RV;1 tumor growth.

Overall, this is an interesting study but does not rise to the impact of an *eLife* publication.

A number of groups have pursed the development of SARD molecules in the last 15+ years and some SARD molecules have been advanced into clinical development. The novelty of this present study is the discovery of a structurally novel SARD molecule with a modestly potent in vitro activity. There is no direct comparison of Z15 with current best SARD molecules to demonstrate any major advantages.

There are some specific deficiencies that need to be addressed.

1. Since Z15 induces AR degradation after 12 h treatment time, the authors should rule out that induced AR degradation is specific. It is possible that the levels of other proteins may also be reduced and reduction of AR protein is one of many mechanisms for the cell growth inhibition of Z15;

2. While Z15 binds to AR LBD with a Kd value of 21 uM, this level of binding could be an artifact, not necessarily a real binding. Therefore, the binding assay needs to be further optimized;

3. Z15 induces degradation of AR in cells and effectively inhibits tumor growth in the 22RV1 xenograft model in mice, AR degradation in tumor tissues needs to be demonstrated.

[Editors' note: further revisions were suggested prior to acceptance, as described below.]

Thank you for resubmitting your work entitled "Selective androgen receptor degrader (SARD) to overcome antiandrogen resistance in castration-resistant prostate cancer" for further consideration by *eLife*. Your revised article has been evaluated by Wafik El-Deiry (Senior Editor) and a Reviewing Editor.

The manuscript has been improved but there are some remaining issues that need to be addressed, as outlined below:

*Reviewer #1 (Recommendations for the authors):*

I thank the authors for their continued work and find the revised manuscript to be much improved. The work was particularly strengthened by the data showing binding of Z15 to both the AR ligand binding domain and the transactivation domain – which makes their other data make much more sense.

I have one significant area where I suggest further improvement. In lines 226-244, the authors describe the LNCaP prostate cancer cell line as castration resistant. I and most investigators generally consider LNCaP cells to be castration sensitive. Conversely, the sub-line of LNCaP, C42B, is generally considered to be castration resistant but retains expression of the androgen receptor.

*Reviewer #3 (Recommendations for the authors):*

The authors have performed additional experiments to address concerns raised in the previous review. Despite their great efforts, the following points are still not being addressed.

1. While it is clear that Z15 reduces the levels of wild-type AR and ARV7, the proposed biochemical mechanism of action is not convincing, The main issue is that Z15 binds to AR LBD with Kd = 21 μm and to AR AF1 domain with Kd = 15 uM, it has a biological activity at sub-μM ranges, much stronger than its direct binding to AR LBD and AR AF1. Based upon its chemical structure, Z15 should have a high plasma protein binding and would require higher concentrations in cells to achieve its biological activity if the proposed direct binding, followed by induced degradation is true. Therefore, it is very likely that Z15 reduces the levels of wild-type AR and ARV7 through a different mechanism, other than those proposed by the authors.

2. While Z15 reduces the levels of wild-type AR and ARV7 proteins in cells and in tumor tissues, it is still unclear if Z15 also reduces the levels of many other proteins in cells and in tumor tissues. The authors have examined three other proteins in the revision and showed that Z15 does not reduce these other three proteins. However, in order to be certain that Z15 specifically reduces the levels of wild-type AR and ARV7 proteins in cells, a more global proteomic analysis would be needed.

[Editors' note: further revisions were suggested prior to acceptance, as described below.]

Thank you for resubmitting your work entitled "Selective androgen receptor degrader (SARD) to overcome antiandrogen resistance in castration-resistant prostate cancer" for further consideration by *eLife*. Your revised article has been evaluated by Wafik El-Deiry (Senior Editor) and a Reviewing Editor.

The manuscript has been improved but there are some remaining issues that need to be addressed, as outlined below:

Please see the comments of reviewer #3 below.

In addition, the manuscript still needs significant English language proofreading due to errors throughout and awkward sentences. Some examples include:

Abstract: "restrict the second-generation antiandrogens benefit"

Lines 47-51: "Clinical studies have shown that the FDA approved second-generation antiandrogens, including AR antagonists (enzalutamide [ENZa], apalutamide and darolutamide) and androgen-synthesizing enzyme inhibitors (abiraterone) benefits patients with CRPC in overall survival time and prostate-specific antigen (PSA) decline"

Lines 51-53: "However, 30~40% of patients with CRPC fail to respond to ENZa or abiraterone, even the remaining patients developed resistance after a few months of the initial response"

Lines 53-56: "To date, multiple possible mechanisms for the resistance development have been identified, among which the mutations in the AR LBD, AR amplification, AR splice variants (ARVs) expression, and intra-tumoral de novo synthesis of androgens had been broadly observed in the clinic"

Line 72-73: "While, current AR PROTACs cannot induce the degradation of ARVs that lack LBD."

Lines 116-118: "In 22Rv1 cells which naturally express full-length AR and ARV7, DHT-induced PSA-luc activity was also dose-dependently blocked by Z15 (Figure 1E)."

Lines 120-122: "We also detected the selectivity of Z15 in GR-positive PC-3 cells co-transfected with the MMTV-luc expression vector plasmid, which were incubated with dexamethasone (Dex) and different concentrations of Z15 for 24 h."

Line 147-148: "We further checked AR and PSA protein levels in LNCaP cells under Z15 and DHT treated by western blot."

Line 159-160: "Next, 4D-lable free proteomics were performed to analysis the influence of Z15 on global protein levels in LNCaP cells."

Line 169-170: "Western blot analysis indicated that Z15 have no influence on GR, HSP90 and CDK7 protein levels in 22Rv1 cells (Figure 2—figure supplement 3)."

Lines 178-181: "As shown in Figure 3A-B, DHT-binding promoted the importing of AR into the nuclear relative to the DMSO-treated group, while both ENZa and Z15 partially reversed the DHT-induced AR nuclear localization."

Line 192-194: "In line with the BLI assay consequence, Z15 (IC50 = 45.32 nM) still exhibited similar AR binding affinity to ENZa (IC50 = 78.95 nM) (Figure 4C)."

Line 201-202: "AR AF1 was previously considered as an important molecule target region of AR."

Line 213-214: "To confirm this hypothesis, we investigated the AR protein and mRNA levels without DHT in LNCaP cells under Z15 treated."

Lines 235-239: "As a result, the proliferation of both VCaP and 22Rv1 cells were significantly inhibited by Z15 in a dose-dependent manner, while Z15 (IC50 = 1.37 μM) showed comparable proliferation inhibition potency with positive control ARV-110 (IC50 = 0.86 μM) in VCaP cells, the 22Rv1 cells proliferation inhibition potency of Z15 (IC50 = 3.63 μM) was greater than ARV-110 (IC50 = 14.85 μM)."

Lines 251-254: "What's more, Z15 treatment also promoted the apoptosis of AR positive CRPC cell lines (VCaP, 22Rv1) but not AR negative DU145 cells both in dose-dependent and time-dependent manners (Figure 6D-E and Figure 6—figure supplement 1A-E)."

Lines 257-259: "Therefore, we designed and optimized the synthesis route of compound Z15 (Figure 7—figure supplement 1) and prepared the gram level of the compound for an in vivo anti-tumor assay in nude mice."

Line 259-260: "Particularly, we investigated the efficacy of Z15 in vivo by 22Rv1 xenografts in male BALB/c nude mice."

Line 260-261: "A total of 5 ×106 22Rv1 cells were injected into the left flank of the male mice and allowed to develop tumors volumes of ~100 mm3."

Lines 268-270: "Western blot analysis indicated that the tumor AR, ARV7, and PSA protein levels were significantly declined in both the 10 and 20 mg/kg Z15 treated group (Figure 7D, Figure 7—figure supplement 2A-C)."

Lines 275-276: "Since Z15 showed inspiring CRPC inhibition potential, we next performed a chemical structure similarity search in SciFinder and the ZINC database to seek Z15 analogs."

Lines 279-283: "As a result, most of these compounds pronouncedly inhibited 280 DHT-induced AR transcriptional activity at a concentration of 1 μM except for ZL-1 (Figure 8—figure supplement 2A), suggesting that the dimethyl isoxazole group played an indispensable role in the AR inhibitory activity of Z15 and its analogs."

Lines 304-306: "Moreover, Z15 could also degrade AR potentially through the proteasome pathway, which supports Z15 to be dual function AR inhibitor and degrader molecule (Figure 9)."

Lines 306-308: "Importantly, several Z15 analogs from the structural similarity search exhibit stronger AR inhibitory and downregulation potency than Z15, which suggest Z15 is a promising lead compound for further optimization."

Line 311-312: "which meant AR antagonizing methods were too faint to overcome the AR

amplification".

Line 334: "but none of these compounds had been gained clinical success".

Please provide a detailed Figure legend for Figure 9 that outlines the mechanism. At present, it is just a one-sentence title for the figure.

*Reviewer #3 (Recommendations for the authors):*

In the revised manuscript, the authors have performed several experiments to confirm the direct binding of Z15 to AR. These new experiments included measuring the binding of Z15 to proteins from cytosolic lysates of LNCaP cells by competing with radio-labeled DHT and the surface plasmon resonance assay, which provide additional evidence that Z15 binds to AR. In addition, the authors have performed global proteomics experiments to show that AR, as well as several AR-regulated gene products, were down-regulated. Although it is still very likely Z15 has a complex cellular mechanism of action, the proposed mechanism action by the authors (i.e. direct binding to AR and induced degradation of AR and ARV7) is responsible in part for the anti-tumor activity of Z15 in prostate cancer in vitro and in vivo. It is recommended that the authors make the following revisions before the publication of the manuscript.

1. Use the binding from the surface plasmon resonance assay as the primary biochemical evidence that Z15 binds to AR, due to the limitations of the BLI assay as the authors indicated in their letter.

2. Provide a Table for all those proteins down-regulated by ARV-110 and Z15 and compare those proteins commonly down-regulated by both compounds, as well as additional proteins, which are differentially regulated by ARV-110 and Z15.

The proteomic data further showed that while AR is obviously the protein most profoundly reduced by ARV-110, AR is only among those proteins reduced by Z15, suggesting that Z15 is a much less specific degrader of AR, as compared to ARV-110. The authors should add discussions to acknowledge that while down-regulation of AR and ARV7 appears to be a major mechanism for the antitumor activity of Z15, Z15 may have other mechanisms of action which contribute to its anticancer activity in vitro and in vivo.

Overall, Z15 is an interesting compound but is not a specific degrader of AR and ARV7. The cellular mechanism of action for Z15 is much more complex than the authors tried to portray.

---

## [Author Response]

Essential revisions:

The data has novel and interesting findings with the small molecule for potential further development in castrate-resistant prostate cancer but the reviewers cautioned against over-interpreting or over-selling the proposed mechanism. The manuscript requires extensive proofreading for english language.Reviewer #1 (Recommendations for the authors):1. Despite proposing that Z15 binds to the AR ligand binding domain, it appears to have similar efficacy against wild type AR and the splice isoform ARv7 which lack the AR ligand binding domain. For example, please see figure 2E. Z15 has similar effects on wild type AR (top band) and ARv7 (bottom band). Multiple high profile publications have reported lack of the ligand binding domain in ARv7. For example, see Antonarakis et al. N Engl J Med 2014; 371:1028-1038.

Thank you very much for this recommendation. ARV7 which lacks the ligand-binding domain but remains constitutively active as a transcription factor, the works of Antonarakis et al and other researches indicated the ARV7 expression was associated with the resistance of CRPC to enzalutamide or abiraterone. As Z15 displays both the full length-AR and the ARV7 inhibition activity, while ARV7 does not possess LBD region, we wonder if Z15 could also bind to other region of AR and then induce ARV7 degradation. The AF1 of AR was considered as an important molecule target region of AR previously, and our BLI assay found that AR AF1 inhibitor UT-34 could bind to AR AF1 directly with KD value of 9.3 μM (Figure 4D), however, UT-34 could not induce ARV7 degradation in 22Rv1 cells in our western blot analysis (Figure 4-figure supplement 2D-F). Interestingly, Z15 was also detected binding to AR AF1 with KD value of 15.0 μM (Figure 4E). Not surprisingly, we didn’t find any binding affinity between AR AF1 and ENZa even at a high concentration of 200 μM (Figure 4F). These data illustrated that Z15 potently inhibits AR-V7 by directly binding to AR AF1. Moreover, that Z15 was also identified binding to the AR LBD. Therefore, Z15 displays the similar efficacy against wild type AR and th AR-V7.

2. The data that Z15 affects AR protein stability as opposed to AR protein expression by some other means is not convincing. For example, see the cyclohexamide experiment in figure 5D. There is minimal difference between the cyclohexamide minus conditions (left half of figure) and the cyclohexamide containing conditions (right half of figure).

Thank you for this suggestion. We performed the densitometric analysis of the band intensities in Figure 5D, and the AR/β-actin quantitative results showed that Z15 significantly accelerated AR degradation in LNCaP cells treated with CHX (right half of Figure 5D) compared to Z15 blank conditions (left half of Figure 5D) (Figure 5-figure supplement 1E).

I think Z15 is a useful molecule, but please do not over-sell the data. I do not think that it acts through your proposed mechanism. However, Z15 is specific to AR expressing cell lines, and efficacious in vivo. I suggest only publishing that much of the story and leaving the downstream mechanism to a subsequent publication.

Thank you for this insightfully suggestion. In the revised manuscript, we tried our best to explain why Z15 induced both AR and ARV7 degradation, and found that the main reason was that Z15 could bind to both LBD and AF1 domain of AR (Figure 4A-4F). While we also think that the exhaustive mechanism about Z15 induced AR and ARV7 positive prostate cancer cells inhibition activity is worth to be explored.

The manuscript is in need of English language editing.

Thank you for the kindly suggestion. The language of the manuscript has been polished by the English writing agency in the revised manuscript

Reviewer #2 (Recommendations for the authors):The manuscript reports the discovery of a novel selective androgen receptor degrader (SARD) targeting the androgen receptor ligand binding domain (AR-LBD) via virtual screening and bioassays. However, the potential targets and binding sites for the degradation of AR and ARVs have not been clarified because the binding of AR-LBD may not induce the degradation of ARVs. For instance, Ponnusamy et al. (10.1158/0008-5472.CAN-17-0976) reported SARD not only binds to AR LBD but also binds to the amino-terminal transcriptional activation domain (AF-1) of AR. Overall, this paper is not suitable to be published on this journal.

Thank you for this insightfully suggestion. We referenced the works of Ponnusamy et al. to investigated the mechanism of Z15 induced ARV7 degradation. As AR AF1 was considered as an important molecule target region of AR previously, and Ponnusamy et al. reported SARD not only binds to AR LBD but also binds to the AR AF1. Our BLI assay found that previous reported AR AF1 inhibitor UT-34 could binding to AR AF1 directly with K_D_ value of 9.3 μM (Figure 4D). By the same method, Z15 was also detected potently binding to AR AF1 with K_D_ value of 15.0 μM (Figure 4E). Not surprisingly, we didn’t find any binding affinity between AR AF1 and ENZa even at a high concentration of 200 μM (Figure 4F). These data illustrated that Z15 potently inhibits ARV7 by directly binding to AR AF1.

1. The authors demonstrate that Z15 is a SARD with glucocorticoid receptor (GR) transcription inhibition activity. The authors should test the transcription inhibition activities of Z15 against progesterone receptor (PR) and mineralocorticoid receptor (MR) due to their high structural similarities.

Thank you for this suggestion. As we found Z15 showed powerful AR transcription inhibition activity at the concentration of 10 μM, while its glucocorticoid receptor (GR) transcription inhibition activity was quite feeble (Figure 1—figure supplement 2B-C). Z15 hardly inhibited dexamethasone (Dex) induced GR transcription activity comparing to the GR antagonist mifepristone (Mif) (Figure 1G). We further compared the steroidal receptor transcription inhibition activity of Z15 in AR, GR, estrogen receptor (ER) and progesterone receptor (PR) dual-luciferase reporter assays, the exogenous AR transcription inhibition IC_50_ of Z15 was 0.41 μM (Figure 1—figure supplement 3A), while both GR and ER transcription inhibition IC_50_ of Z15 were over 20 μM (Figure 1—figure supplement 3B-C), the PR transcription inhibition IC_50_ of Z15 was 9.29 μM (Figure 1—figure supplement 3D), these results suggested that Z15 was a highly selective AR inhibitor.

2. Lines 110-111. The authors state 'The results indicated that Z15 could inhibit DHT-induced transcriptional activities of both exogenous and endogenous AR in a dose-dependent manner (Figure 1B, 1C, 1D).' However, it seems Z15 inhibits VCaP not in a dose-dependent manner (Figure 1D).

Thank you for this suggestion. We changed the description in the revised manuscript as: To further test the AR inhibition potency of Z15, we next performed dual-luciferase reporter assay in several human prostate cancer cell lines including wt-AR-transfected PC-3, LNCaP, and 22Rv1 cells. The results indicated that Z15 could inhibit DHT-induced transcriptional activities of both exogenous and endogenous AR in a dose-dependent manner (Figure 1B-C). In VCaP cells which overexpress AR, we found that 5 μM enzalutamide (ENZa) could not inhibit AR transcription activity, while Z15 showed potently AR transcription inhibition activity even in low concentration (Figure 1D).

3. The authors describe and interpret their data using the word 'significantly' for the Western blot result. Quantitative results for all Western blot results in the manuscript should be presented, such as using the densitometric analysis of the band intensities.

Many thanks for the suggestion. the densitometric analysis of the band intensities for all Western blot results in the revised manuscript were presented in supplementary materials (Figure 2—figure supplement 1A-I. Figure 4—figure supplement 2B-C, E-F. Figure 5—figure supplement 1A-G. Figure 6—figure supplement 1C-D. Figure 7—figure supplement 2A-C. Figure 8—figure supplement 3A-D).

4. The authors should quantitatively analyze the AR nuclear localization results in Figure 3.

Thank you for this suggestion. The quantitatively analysis of the AR nuclear localization results in Figure 3A was presented in Figure 3B.

5. Positive control should be added in Figure 6A-F, such as Enzalutamide.

Thank you for this suggestion. As AR LBD targeted degrader ARV-110 showed potently prostate cancer inhibition activity in clinical trials, we chose ARV-110 as positive control to analysis whether Z15 induced AR inhibition and degradation could lead to the CRPC cell growth inhibition. We found that the 22Rv1 cells proliferation inhibition potency of Z15 (IC_50_ = 3.63 μM) was greater than ARV-110 (IC_50_ = 14.85 μM) (Figure 6A), Z15 strongly decreased 22Rv1 cells colony numbers compared to both untreated group and ARV-110 treated group (Figure 6B), Z15 treatment also showed the stronger apoptosis induction activity of AR positive CRPC cell lines (LNCaP, 22RV1) than ARV-110 (Figure 6D-E).

6. The loading controls (β-action) are uneven, indicating the samples are not quantified before Western blot analysis

Thank you for this comment. We checked all the loading controls (β-action) presenting in our manuscript, we found some loading controls (β-action) truly are uneven, such as Figure 5D, Figure 8C, as we performed densitometric analysis of the band intensities for all Western blot results in the revised manuscript (Figure 2—figure supplement 1A-I. Figure 4—figure supplement 2B-C, E-F. Figure 5—figure supplement 1A-G. Figure 6—figure supplement 1C-D. Figure 7—figure supplement 2A-C. Figure 8—figure supplement 3A-D), we think this problem could not influence our judgment to the Western blot results.

Reviewer #3 (Recommendations for the authors):The present study reports the discovery and preclinical evaluation of a novel therapeutic agent for the treatment of castration-resistance prostate cancer through inducing degradation of androgen receptor.The major strengths of this study is the identification of a novel lead compound and its interesting in vitro and in vivo activities in prostate cancer models. The major weaknesses with this study are that since the concept of induced AR degradation is not new, additional data are needed to demonstrate some major advantages of the agent reported by the authors over current AR degraders and its potential for clinical translation.

Thank you for this insightfully suggestion. Based on our data presented in the revised manuscript, there are some advantages of Z15 compared to current reported AR degraders. Compared to AR PROTAC ARV-110, Z15 could not only bind to AR LBD but also AR AF1 (Figure 4A-4F), thus that ARV7 positive enzalutamide resistant 22Rv1 cells proliferation inhibition potency of Z15 (IC50 = 3.63 μM) was greater than ARV-110 (IC50 = 14.85 μM) (Figure 6A), Z15 strongly decreased 22Rv1 cells colony numbers compared to both untreated group and ARV-110 treated group (Figure 6B), Z15 treatment also showed the stronger apoptosis induction activity of AR positive CRPC cell lines (LNCaP, 22RV1) than ARV-110 (Figure 6D-E). Compared to AR AF1 inhibitor UT-34 which was reported possess ARV7 degradation activity, Z15 has similar mechanism, as the AR DC50 of Z15 in LNCaP cells was 1.05 μM (Figure 2G, Figure 2-figure supplement 1H), in 22Rv1 cells, the AR DC50 was 1.16 μM, the ARV7 DC50 was 2.24 μM (Figure 2H, Figure 2-figure supplement 1I), however, UT-34 could not induce ARV7 degradation in 22Rv1 cells in our western blot analysis (Figure 4-figure supplement 2D-F), which indicated that Z15 may has stronger ARV7 degrade activity than UT-34. Based on PCa patient tumor tissues, we further cultured PCa organoids and treated organoids with 1 μM Z15 for 7 days. The results showed that Z15 significantly inhibited PCa organoid proliferation compared to the control group (Figure 6C), which showed the potential of Z15 for clinical translation.

The authors have identified a good lead compound but further optimization and studies are needed to make a true impact for the filed of developing AR degraders for the treatment of castration-resistance prostate cancer.

Thank you for this suggestion. Seven compounds (structures shown in Figure 8-figure supplement 1) with more than 80% similarity with Z15 were finally purchased for the further bioactivity evaluation. ZL-2 and ZL-4 exhibited the most AR transcription inhibitory potency, while others also showed comparable AR inhibition activity compared to Z15 (Figure 8A-B). In addition, Western blot analysis revealed that these active molecules could reduce AR and PSA protein levels in a dose-dependent manner. Notably, most of them showed stronger AR down-regulation activity than Z15 (Figure 8C, Figure 8-figure supplement 3A-D). Together, these active Z15 analogs indicate that through the chemical structural modification of Z15, we may find out more novel selective androgen receptor degraders with better AR inhibition activity in a shorten future. Moreover, we have designed the synthesis route of Z15, and prepared more than one gram of product for the in-vivo assay. It would be conveniently to expand the compound space to optimize Z15 via the design route.

This study reported identification of a novel selective androgen receptor degrader molecule (SERD), which the authors named Z15. The authors have provided data to demonstrate the following for Z15:1) inducing degradation of wild-type AR and AR-V7 in a proteasome-dependent manner;2) displaying fairly potent cell growth inhibition of AR+ LNCaP, VCaP and 22RV1 cell lines and some selectivity over AR- PC-3 and DU-145 cell line;3) binding to AR ligand binding domain (LBD) directly with a Kd value of 21 μm and blocking the binding of AR agonist with AR LBD;4) inhibiting 22RV;1 tumor growth.Overall, this is an interesting study but does not rise to the impact of an eLife publication.A number of groups have pursed the development of SARD molecules in the last 15+ years and some SARD molecules have been advanced into clinical development. The novelty of this present study is the discovery of a structurally novel SARD molecule with a modestly potent in vitro activity. There is no direct comparison of Z15 with current best SARD molecules to demonstrate any major advantages.

Thank you for this suggestion. The most interesting properties for Z15 are that Z15 could inhibit both FL-AR and AR-V7, as well as its degradation potency for both FL-AR and AR-V7. Based on our data presented in the revised manuscript, there are some advantages of Z15 compared to current reported AR degraders. Compared to AR PROTAC ARV-110, Z15 could not only bind to AR LBD but also AR AF1 (Figure 4A-F), thus that ARV7 positive enzalutamide resistant 22Rv1 cells proliferation inhibition potency of Z15 (IC_50_ = 3.63 μM) was greater than ARV-110 (IC_50_ = 14.85 μM) (Figure 6A), Z15 strongly decreased 22Rv1 cells colony numbers compared to both untreated group and ARV-110 treated group (Figure 6B), Z15 treatment also showed the stronger apoptosis induction activity of AR positive CRPC cell lines (LNCaP, 22RV1) than ARV-110 (Figure 6D-E). Compared to AR AF1 inhibitor UT-34 which was reported possess ARV7 degradation activity, Z15 has similar mechanism, as the AR DC50 of Z15 in LNCaP cells was 1.05 μM (Figure 2G, Figure 2—figure supplement 1H), in 22Rv1 cells, the AR DC50 was 1.16 μM, the ARV7 DC50 was 2.24 μM (Figure 2H, Figure 2—figure supplement 1I), however, UT-34 could not induce ARV7 degradation in 22Rv1 cells in our western blot analysis (Figure 4—figure supplement 2), which indicated that Z15 may has stronger ARV7 degrade activity than UT-34. Based on PCa patient tumor tissues, we further cultured PCa organoids and treated organoids with 1 μM Z15 for 7 days. The results showed that Z15 significantly inhibited PCa organoid proliferation compared to the control group (Figure 6C), which showed the potential of Z15 for clinical translation.

There are some specific deficiencies that need to be addressed.1. Since Z15 induces AR degradation after 12 h treatment time, the authors should rule out that induced AR degradation is specific. It is possible that the levels of other proteins may also be reduced and reduction of AR protein is one of many mechanisms for the cell growth inhibition of Z15;

Thank you for this suggestion. To verify the specificity of Z15 downregulated AR protein levels, we chose 3 AR pathway related but independent proteins GR, HSP90 (AR chaperonin) and cyclin-dependent kinases 7 (CDK7) as controls. Western blot analysis indicated that Z15 have no influence on GR, HSP90 and CDK7 protein levels in 22Rv1 cells (Figure 2—figure supplement 2)

2. While Z15 binds to AR LBD with a Kd value of 21 uM, this level of binding could be an artifact, not necessarily a real binding. Therefore, the binding assay needs to be further optimized;

Thank you for this comment. According to the manual of ForteBio Octet RED96 instrument (ForteBio, Inc, CA, USA), the ForteBio Octet detected protein-small molecule (MW < 2 kDa) general affinity (K_D_) is about 10^-4^ – 10^-7^ M, thus, we think that the ForteBio Octet detected binding affinity between Z15 and AR LBD with a K_D_ Value of 21 μM is generally convincing. Moreover, the competitive AR LBD binding assay was performed to demonstrate the direct interaction between Z15 and AR LBD. Synthetic androgen R1881 displayed strong binding potency to AR with IC_50_ value of 0.30 nM, which indicated this assay system's feasibility. In line with the BLI assay consequence, Z15 (IC_50_ = 45.32 nM) still exhibited similar AR binding affinity to ENZa (IC_50_ = 78.95 nM) (Figure 4C). Besides, our fluorescence polarization (FP) assay also demonstrated Z15 could compete with androgen binding to AR LBD (Figure 4—figure supplement 1). These data suggested that Z15 could antagonize AR by targeting the ligand binding domain directly.

3. Z15 induces degradation of AR in cells and effectively inhibits tumor growth in the 22RV1 xenograft model in mice, AR degradation in tumor tissues needs to be demonstrated.

Thank you for this great suggestion. We have repeated the in vivo assay and detected similar tumor growth inhibition activity of Z15, also, AR and PSA protein levels in tumor tissues were detected by Western blot, the results indicated that the tumor AR, ARV7 and PSA protein levels were significantly declined in both 10 mg/kg and 20 mg/kg Z15 dose treated group (Figure 7D, Figure 7—figure supplement 2A-C).

[Editors' note: further revisions were suggested prior to acceptance, as described below.]

The manuscript has been improved but there are some remaining issues that need to be addressed, as outlined below:Reviewer #1 (Recommendations for the authors):I thank the authors for their continued work and find the revised manuscript to be much improved. The work was particularly strengthened by the data showing binding of Z15 to both the AR ligand binding domain and the transactivation domain – which makes their other data make much more sense.I have one significant area where I suggest further improvement. In lines 226-244, the authors describe the LNCaP prostate cancer cell line as castration resistant. I and most investigators generally consider LNCaP cells to be castration sensitive. Conversely, the sub-line of LNCaP, C42B, is generally considered to be castration resistant but retains expression of the androgen receptor.

Thank you very much for this recommendation. LNCaP should exactly be a castration sensitive cell line. As we have some difficulties in purchasing or asking C4-2B cells from ATCC or other labs, we chose VCaP cells for further investigation which are also generally considered to be castration resistant but retains expression of the androgen receptor. As a result of MTT assay, the proliferation of VCaP cells was significantly inhibited by Z15 in a dose-dependent manner, Z15 (IC_50_ = 1.37 μM) showed comparable proliferation inhibition potency with positive control ARV-110 (IC_50_ = 0.86 μM) in VCaP cells (Figure 6A). What’s more, Z15 treatment also promoted the apoptosis of VCaP cells (Figure 6D and Figure 6—figure supplement 1A, 1C).

Reviewer #3 (Recommendations for the authors):The authors have performed additional experiments to address concerns raised in the previous review. Despite their great efforts, the following points are still not being addressed.1. While it is clear that Z15 reduces the levels of wild-type AR and ARV7, the proposed biochemical mechanism of action is not convincing, The main issue is that Z15 binds to AR LBD with Kd = 21 μm and to AR AF1 domain with Kd = 15 uM, it has a biological activity at sub-μM ranges, much stronger than its direct binding to AR LBD and AR AF1. Based upon its chemical structure, Z15 should have a high plasma protein binding and would require higher concentrations in cells to achieve its biological activity if the proposed direct binding, followed by induced degradation is true. Therefore, it is very likely that Z15 reduces the levels of wild-type AR and ARV7 through a different mechanism, other than those proposed by the authors.

Thank you very much for this recommendation. LNCaP should exactly be a castration sensitive cell line. As we have some difficulties in purchasing or asking C4-2B cells from ATCC or other labs, we chose VCaP cells for further investigation which are also generally considered to be castration resistant but retains expression of the androgen receptor. As a result of MTT assay, the proliferation of VCaP cells was significantly inhibited by Z15 in a dose-dependent manner, Z15 (IC_50_ = 1.37 μM) showed comparable proliferation inhibition potency with positive control ARV-110 (IC_50_ = 0.86 μM) in VCaP cells (Figure 6A). What’s more, Z15 treatment also promoted the apoptosis of VCaP cells (Figure 6D and Figure 6—figure supplement 1A, 1C).

2. While Z15 reduces the levels of wild-type AR and ARV7 proteins in cells and in tumor tissues, it is still unclear if Z15 also reduces the levels of many other proteins in cells and in tumor tissues. The authors have examined three other proteins in the revision and showed that Z15 does not reduce these other three proteins. However, in order to be certain that Z15 specifically reduces the levels of wild-type AR and ARV7 proteins in cells, a more global proteomic analysis would be needed.

Thank you for this comment. To analysis whether Z15 downregulate AR and ARV7 through directly binding to AR, we performed 4 assays to investigate this problem.

1) The competitive AR binding assay was performed to demonstrate the direct interaction between Z15 and AR LBD, whereby compounds in competition with the radioligand [^3^H] DHT in cytosolic lysates from LNCaP cells were measured. Synthetic androgen R1881 displayed strong binding potency to AR with an IC_50_ value of 0.30 nM, which indicated the feasibility of this assay system. Z15 (IC_50_ = 45.32 nM) exhibited similar AR binding affinity to ENZa (IC_50_ = 78.95 nM) (Figure 4C). Besides, our fluorescence polarization assay also demonstrated Z15 could compete with androgen binding to AR LBD (Figure 4—figure supplement 1).

2) In this revised version, we performed surface plasmon resonance (SPR) assay to detect binding affinity between Z15 and AR AF1, the results showed that the KD value was 0.93 μM (Figure 4—figure supplement 3). This KD value was more reasonable, as the ARV7 DC_50_ of Z15 was about 2.24 μM.

3) The KD value detected by our BLI assay were much higher, which Z15 binds to AR LBD with Kd = 21 µM and to AR AF1 domain with Kd = 15 µM. However, the KD value between Z15 and AR LBD or Z15 and AR AF1 were comparable to the KD value between positive control ENZa for AR LBD or UT-34 for AR AF1(Figure 4A-B, D-F). We also consulted the engineer of Sartorius Group (instrument supplier) about this problem, we known that ForteBio Octet RED96 (our BLI analysis instrument) is more suitable for protein-protein interaction analysis, the protein-compound analysis by this instrument may less sensitive, so, positive control is very important. This could partially explain why the biological activity at sub-μM ranges of Z15, much stronger than its direct binding to AR LBD and AR AF1.

4) We performed 4D-lable free proteomics analysis, KEGG analysis proved that Z15 and AR LBD directly binding PROTAC molecule ARV-110 had similar influence on functional pathways of LNCaP cells (Figure 2—figure supplement 2C-D), which indicated that Z15 reduces AR protein levels unlikely through other pathways. However, we also found Z15 treated group showed more stronger inhibition activity of lipid metabolism related proteins and pathways compared to ARV-110 group. We reasoned this phenomenon partially induced by the down regulation of SREBF1 which is an AR downstream regulate protein.

Therefore, these results trend to support that Z15 downregulate AR and ARV7 through directly binding to AR LBD and AR AF1.

[Editors' note: further revisions were suggested prior to acceptance, as described below.]

The manuscript has been improved but there are some remaining issues that need to be addressed, as outlined below:Please see the comments of reviewer #3 below.Reviewer #3 (Recommendations for the authors):In the revised manuscript, the authors have performed several experiments to confirm the direct binding of Z15 to AR. These new experiments included measuring the binding of Z15 to proteins from cytosolic lysates of LNCaP cells by competing with radio-labeled DHT and the surface plasmon resonance assay, which provide additional evidence that Z15 binds to AR. In addition, the authors have performed global proteomics experiments to show that AR, as well as several AR-regulated gene products, were down-regulated. Although it is still very likely Z15 has a complex cellular mechanism of action, the proposed mechanism action by the authors (i.e. direct binding to AR and induced degradation of AR and ARV7) is responsible in part for the anti-tumor activity of Z15 in prostate cancer in vitro and in vivo. It is recommended that the authors make the following revisions before the publication of the manuscript.1. Use the binding from the surface plasmon resonance assay as the primary biochemical evidence that Z15 binds to AR, due to the limitations of the BLI assay as the authors indicated in their letter.

Thank you for this kind comment. We have redrawn Figure 4 and Figure 4—figure supplement 2, the AR competitive binding assay result was used as primary biochemical evidence that Z15 binds to AR LBD (Figure 4A), the surface plasmon resonance assay result was used as primary biochemical evidence that Z15 binds to AR AF1 (Figure 4C). The BLI assay results were mainly shown in supplementary materials (Figure 4—figure supplement 2).

2. Provide a Table for all those proteins down-regulated by ARV-110 and Z15 and compare those proteins commonly down-regulated by both compounds, as well as additional proteins, which are differentially regulated by ARV-110 and Z15.

Thank you for this helpful suggestion. Supplementary file 1d and Supplementary file 1e were re-arrangement to list all those proteins down-regulated by ARV-110 and Z15. The proteomic data reveals that there are 19 proteins commonly down-regulated by both ARV-110 and Z15, these proteins account for 55.9% (19 among 34) in ARV-110 down-regulated proteins (Supplementary file 1d) and 27.5% (19 among 69) in Z15 down-regulated proteins (Supplementary file 1e). We think these two re-arrangement tables showed ARV-110 and Z15 down-regulation proteins situation more clearly to the readers.

3. The proteomic data further showed that while AR is obviously the protein most profoundly reduced by ARV-110, AR is only among those proteins reduced by Z15, suggesting that Z15 is a much less specific degrader of AR, as compared to ARV-110. The authors should add discussions to acknowledge that while down-regulation of AR and ARV7 appears to be a major mechanism for the antitumor activity of Z15, Z15 may have other mechanisms of action which contribute to its anticancer activity in vitro and in vivo.

Thank you for your insightful suggestion, this suggestion combined with the previous recommendation about the mechanisms of action of Z15 gives us many inspirations, and we are very grateful for this. In the revised manuscript, we discussed that Z15 may have other mechanisms of action which contribute to its anticancer activity in vitro and in vivo. It is described as follows:

“However, we notice that the AR degradation potency of Z15 is not as stronger as AR-targeted PROTAC molecule ARV-110, suggesting that Z15 is a less specific AR degrader compared to ARV-110. We acknowledge that while Z15 suppresses CRPC progress mainly through targeting AR and ARV7, Z15 may have other mechanisms of action which contribute to its anticancer activity in vitro and in vivo. Nevertheless, Z15 is a novel selective AR and ARV7 degrader that warrants further structure optimization and anti-CRPC mechanisms investigation.”